



# Epistemic uncertainties and natural hazard risk assessment. 1. A review of different natural hazard areas

Keith J. Beven[1,2], Susana Almeida[3], Willy P. Aspinall[4], Paul D. Bates[5], Sarka Blazkova[6], Edoardo Borgomeo[7], Katsu Goda[3], Jim W. Hall[7], Jeremy C. Phillips[4], Michael Simpson[7], Paul J. Smith[1,8], David B. Stephenson[9], Thorsten Wagener[3,10], Matt Watson[4], and Kate L. Wilkins[4]

[1] Lancaster Environment Centre, Lancaster University, Lancaster, UK

[2] Department of Earth Sciences, Uppsala University, Uppsala, Sweden

[3] Department of Civil Engineering, Bristol University, Bristol, UK

[4] School of Earth Sciences, Bristol University, Bristol, UK

[5] School of Geographical Sciences, Bristol University, Bristol, UK

[6] T. G. Masaryk Water Resource Institute, Prague, Czech Republic

[7] Environmental Change Institute, Oxford University, UK

[8] European Centre for Medium-Range Weather Forecasting, Reading, UK

[9] Department of Mathematics and Computer Science, Exeter University, Exeter, UK

[10] Cabot Institute, University of Bristol, UK

*Correspondence to*: Keith J. Beven (k.beven@lancaster.ac.uk)

**Abstract:** This paper discusses how epistemic uncertainties are considered in a number of different natural hazard areas including floods, landslides and debris flows, dam safety, droughts, earthquakes, tsunamis, volcanic ash clouds and pyroclastic flows, and wind storms. In each case it is common practice to treat most uncertainties in the form of aleatory probability distributions but this may lead to an underestimation of the resulting uncertainties in assessing the hazard, consequences and risk. It is suggested that such analyses might be usefully extended by looking at different scenarios of assumptions about sources of epistemic uncertainty, with a view to reducing the element of surprise in future hazard occurrences. Since every analysis is necessarily conditional on the assumptions made about the nature of sources of epistemic uncertainty it is also important to follow the guidelines for good practice suggested in the companion Part 2.

## 1 Introduction

With the increasing appreciation of the limitations of traditional deterministic modelling approaches, uncertainty estimation has become an increasingly important part of natural hazards assessment and risk management. In part, this is a natural




extension of the evaluation of frequencies of hazard in assessing risk, in part an honest recognition of the limitations of any risk analysis, and in part because of the recognition that most natural hazards are not stationary in their frequencies of occurrence.

5   The consideration of uncertainty in risk assessments has, however, been relatively uncommon.   Figure 1 shows some statistics on the publication of papers concerned with uncertainty assessment for different types of natural hazards.   While these show an increase over the past 15 years, some hazard types return zero results, and a search on "epistemic uncertainty" AND "natural hazards" returned zero results on Web of Science.

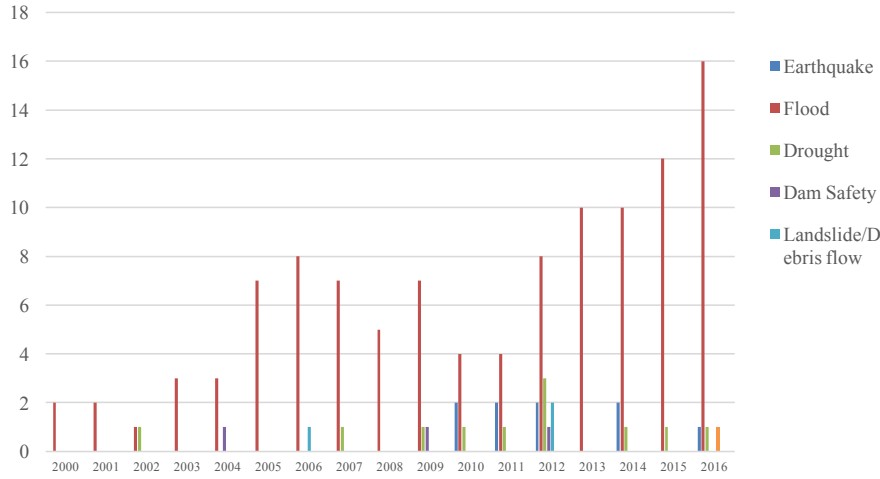

**Figure 1:   Numbers of papers published since the year 2000 according to a search on Web of Science using "uncertainty estimation" together with various natural hazard descriptors.   Note that several return zero results.**

A large number of natural hazards can be framed in a Hazard-Magnitude-Footprint-Loss setting (e.g. Rougier et al., 2013).

15   In this review we discuss the impact of epistemic uncertainty on risk assessments for different types of natural hazards. Throughout, we believe it is important to think about the risk, not just the hazard, which includes thinking about the loss (which is stakeholder specific).   This means that any risk assessment involves a modelling cascade, each element of which involves epistemic uncertainties, with the potential for the uncertainty in risk to grow, or be constrained by additional data, with each component in the cascade (e.g. Beven and Lamb, 2014).

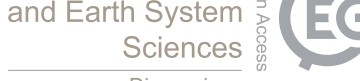



The analysis of risk has, in the past, commonly depended on assessing the probabilities of magnitudes, footprints and loss. Such a probabilistic analysis leads naturally to the use of risk-based decision making in the management of the risk, in terms of either mitigation measures or insurance decisions, where this is possible for different natural hazards.   Probabilistic risk

analysis of this type assumes that the different sources of uncertainty can, at some fundamental level, be treated as random or aleatory variables (and that all possible futures have been considered so that the probability assessments can be taken as complete).

There is, however, an increasing appreciation that this is not the only type of uncertainty that arises in the assessment of

natural hazards (Hoffman and Hammonds, 1994; Helton and Burmaster, 1996; Walker et al., 2003; Brown, 2004, 2010; van der Sluijs et al., 2005; Refsgaard et al., 2006, 2007, 2013; Beven, 2009, 2012, 2013, 2016; Warmink et al., 2010; Rougier and Beven, 2013; Beven and Young, 2013).   In particular, since the time of Keynes (1921) and Knight (1921) it has been common practice to distinguish between those uncertainties that might be represented as random chance, and those, which arise from a lack of knowledge about the nature of the phenomenon being considered.   Knight (1921) referred to the latter as

the "real uncertainties" and they are now sometimes called "Knightian uncertainties".    While Knight's thinking pre-dated modern concepts and developments in probability theory  (e.g. de Finetti, 1937, 1974;  Cox, 1946, and others), the distinction between uncertainties that can be treated simply as probabilistic and additional knowledge uncertainties holds.

In fact, an argument can be made that all sources of uncertainty can be considered as a result of not having enough

knowledge about the particular hazard occurrence being considered: it is just that some types of uncertainty are more acceptably represented in terms of probabilities than others.   In current parlance, there are the "aleatory uncertainties" (from the Latin *"alea"*, meaning a die or game of dice),  while the Knightian real uncertainties are the "epistemic uncertainties" (from the Greek "*ἐπιστήμη*", for knowledge or science).   Aleatory uncertainties represent variability, imprecision and randomness, or factors that can be modelled as random for practical expediency, which can be represented as forms of noise

within a statistical framework. Within epistemic uncertainties it is possible to subsume many other uncertainty concepts such as ambiguity, reliability, vagueness, fuzziness, greyness, inconsistency and surprise that are not easily represented as probabilities. This distinction is important because most methods of decision-making used in risk assessments are based on the concept of risk as the product of a probability of occurrence of an event (the hazard, magnitude and footprint components in the model cascade) and an evaluation of the consequences of that event (the loss component).  If there are important

uncertainties in the assessment of the occurrence that are not easily assessed as probabilities, or if there are significant epistemic uncertainties about the consequences, then some other means of assessing risk decisions might be needed.  Given lack of knowledge, there is also plenty of opportunity for being wrong about the assumptions used to describe sources of uncertainty, or having different belief systems about the representations of uncertainties, which is sometimes referred to as ontological uncertainty  (from the Greek "*ὄνgen*" meaning "being or that which is", the present participle of the verb "*εἰμί*",





to be e.g. Marzocchi and Jordan, 2014; Beven, 2015). Epistemic and ontological uncertainties are also sometimes referred to as "deep uncertainties", including in risk analysis and natural hazards (e.g. Cox, 2012; Stein and Stein, 2013).

For the practical purposes of this review, we will define epistemic uncertainty as those uncertainties that are not well determined by historical observations. This lack of determination can be because the future is not expected to be like the past or because the historical data are unreliable (imperfectly recorded, estimated from proxies, or missing), because they are scarce (either because measurements are not available at the right scale or there is simply no observational network available), because the structure of that uncertainty does not have a simple probabilistic form, or because we expect the probability estimates to be incomplete (unbounded or indeterminable, e.g. Brown, 2004).

In what follows we consider the sources and impact of epistemic uncertainties in different natural hazard areas. We see the typical audience of this opinion piece as a natural hazard scientist who is likely aware of uncertainties in his/her own specific hazard area, while having limited understanding of other hazard areas and of the approaches available to deal with epistemic uncertainties. Obviously it is difficult to go into great detail on each aspect covered here; hence the focus is on providing an

overview and on citing key literature. In the second part of the paper we discuss the different opinions about the options for addressing epistemic uncertainty and we discuss open problems for implementing these options in terms of what might constitute good practice.

## 2 Floods

### 2.1 Floods and key epistemic uncertainties

Floods account for about one third of all economic losses from natural hazards globally (UNISDR, GAR 2015). The frequency and magnitude of flood disasters is likely to increase with a warming atmosphere due to climate change and with increased exposure of a growing population (Winsemius et al., 2015), which suggests that the fractional contribution to global disaster losses is likely to increase even further. There are five aspects of flood risk assessment that involve important

epistemic uncertainties. The first is the assessment of how much rainfall or snowmelt input occurs (either in past or future events); the second is the frequency with which such events might occur and how that might be changing; the third is how much of that input becomes flood runoff; the fourth is the footprint of the flood inundation; and the fifth is the assessment of either past or potential damages (see discussion in Section 11 below). These all apply in the assessment of expected damages for events of different magnitude for making decisions in managing the flood risk and in the management of flood

incidents in real time (e.g. Sayers et al., 2002).

### 2.2 Uncertainty quantification in flood hazard estimation





In the context of flooding, uncertainties in inputs and runoff generation are often avoided by estimating the probability of exceedance for different magnitudes of event in terms of an extreme value distribution of discharges. That does not mean that such uncertainties are not important (such as lack of knowledge about the effects of a poorly known spatial pattern of inputs on runoff generation, the role of antecedent conditions in controlling runoff generation, or estimates of historical flood

peak discharges), only that they are assumed to contribute to some underlying statistical distribution of events that is fitted to the available historical data. That provides estimates of frequency as if the series of historical floods is drawn from a stationary distribution, which is not easily modified to allow for future change (e.g. Prudhomme et al., 2010).

The epistemic uncertainty then is convolved into a question of what statistical distribution should be used. This question has

often been resolved by institutionalising the uncertainty into a particular choice of standard distribution. Different countries have chosen different distributions and, in some cases, have changed that choice over time. There are good theoretical reasons to choose the Generalised Extreme Value (GEV) distribution. Asymptotically a sample of extremes with independent occurrences in successive time periods (e.g. years) from an arbitrary underlying distribution of events should have the form of the GEV distribution. It was the distribution of choice for the analysis of annual maximum floods in the

UK Flood Studies Report (NERC, 1975). However, the time series available for the analysis of floods are often relatively short, so the asymptotic condition may not be approached and the occurrences of events may not be independent in time or space (eg. Eastoe and Tawn, 2010; Keef et al., 2013). Thus in revising the UK methodology in the Flood Estimation Handbook, a change was made to recommend the Generalised Logistic Distribution since it resulted in fewer sites being assigned parameters that suggested some upper limit to flood magnitudes (IH 1999). Many other distributions have been

used elsewhere. A recent development in flood risk management has been a concern with the joint occurrences of flood events, rather than looking at individual sites independently. This requires specifying not only one distribution but joint distributions and the correlation structure between them (e.g. Keef et al., 2013), but which may not be well defined by historical data.

The choice of a particular distribution essentially controls the form of the upper tail of the distribution and consequently the assessment of risk. This is common to the other natural hazards that are considered below. Good practice suggests that the statistical uncertainty associated with the tail of the fitted distribution should be evaluated (although this is rarely reported even where it is provided by the analysis software) but essentially we have additional epistemic uncertainties as to what distribution to choose and whether to treat that distribution as stationary or whether clusters of events might come from some

more complex stochastic structure (e.g. Koutsoyiannis, 2003, 2010; Montanari and Koutsoyiannis, 2012). If this is the case, then it might result in a significant increase in the range of uncertainty relative to classical statistical analysis (e.g. Koutsoyiannis and Montanari, 2007) irrespective of other sources of epistemic uncertainty.

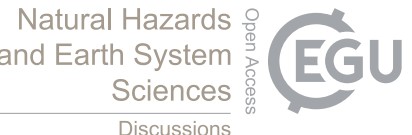



These issues have led some people to step back to considering the inputs and runoff generation over a catchment more directly in flood risk estimation. This approach was pioneered by Eagleson (1972) using a simple derived distribution model of runoff generation, but increased computer power has allowed continuous simulation over long periods of time using rainfall-runoff models which has the advantage that the variation in antecedent wetness of a catchment prior to an

event is part of the simulation (e.g. Beven, 1987; Cameron et al. 1999, 2000; Lamb and Kay, 2004; Blazkova and Beven, 2004, 2009). In some cases it is possible to use long series of observed rainfall data to simulate discharges; but for the very long series that are needed to estimate more extreme events it is necessary to use a stochastic model of the inputs (similar to the weather generators used to produce future sequences in climate change impact assessments). However, this only shifts the epistemic uncertainty issue of the choice of appropriate distributions or more complex stochastic structures for the space-

time characteristics of rainfall (e.g. Chandler et al., 2014). The extreme events generated from such a weather generator depend on the tails of the assumed distribution(s) and there will again be epistemic uncertainty about what type of distribution to use, even where rainfall series are longer than discharge records.

A further advantage of the continuous simulation approach is that the weather generator can be modified to represent future

climates (e.g. Cameron et al., 2000; Wilby and Dessai, 2010; Prudhomme and Davies, 2009; Prudhomme et al., 2010), and that input data might be more readily available for sites for which there are no discharge records (the prediction in ungauged basins problem, Wagener et al., 2004; Blöschl et al., 2013; Hrachowitz et al., 2013). This latter case still requires that the parameters of a rainfall-runoff model be specified. This is also an epistemic uncertainty issue, even if extrapolations from gauged sites are often made using statistical regression or pooling group methods (e.g. Lamb and Kay, 2004) a process that

will be influenced by model structural uncertainty and other uncertainty sources (e.g. McIntyre et al., 2005; Wagener and Wheater, 2006). Experience in predicting the flood characteristics in this way has been somewhat mixed; successful in some basins, but with significant over or underestimation in others (Lamb and Kay, 2004; Blöschl et al., 2013). Improvements to such methods might still be possible but epistemic uncertainty will remain a constraint on accuracy.

Further uncertainties arise in the estimation of the footprint of the flood event. There may be different areas at risk of inundation according to whether the risk is from pluvial, fluvial, coastal or groundwater flooding. By making assumptions about various sources of uncertainty in the modelling of inundation, a forward uncertainty analysis can be used to predict uncertainties in inundation areas and depths using Monte Carlo simulation methods (e.g. Berry et al., 2008). In some cases, historical flood mapping is available that can be used to condition hydraulic models of inundation and constrain the

uncertainty in model predictions (Bates et al., 2014). Both Generalised Likelihood Uncertainty Estimation (GLUE; Aronica et al., 1998; Romanowicz and Beven, 2003; Pappenberger et al., 2007; Neal et al., 2013; Beven et al., 2014; Beven and Lamb, 2014) and more formal Bayesian methods (Romanowicz et al., 1996; Hall et al., 2011) have been used in this type of conditioning process (e.g. Figure 2; see also other examples in Beven et al., 2014) but the results will depend on how the





various model runs are evaluated (given that model residuals are likely to be correlated in space and time) as well as what type of inundation model is being used.

Recent improvements in flood inundation modelling, however, have been less a result of reducing uncertainties in inputs and
5   conveyance parameters, but rather due to the much better definition of flood plain topography as LIDAR surveys have become more widely available.  Even LIDAR however cannot identify all the barriers to flow on a flood plain (e.g. Sampson et al., 2012), and therefore we should expect some interaction between effective conveyance parameters and other features of model implementation in matching historical flood data.   Even then there is some suggestion that the effective conveyance parameters identified for one magnitude of event, might not hold for a larger magnitude event (e.g. Romanowicz and Beven,
10   2003) so that the simple assumption that conveyance parameters are constant might introduce epistemic uncertainty.   It is also common to assume that the effective conveyance parameters are spatially constant which, when interacting with other sources of uncertainty might mean that it is not possible to get good fits to inundation observations everywhere in the modelled domain (e.g. Pappenberger et al., 2007).

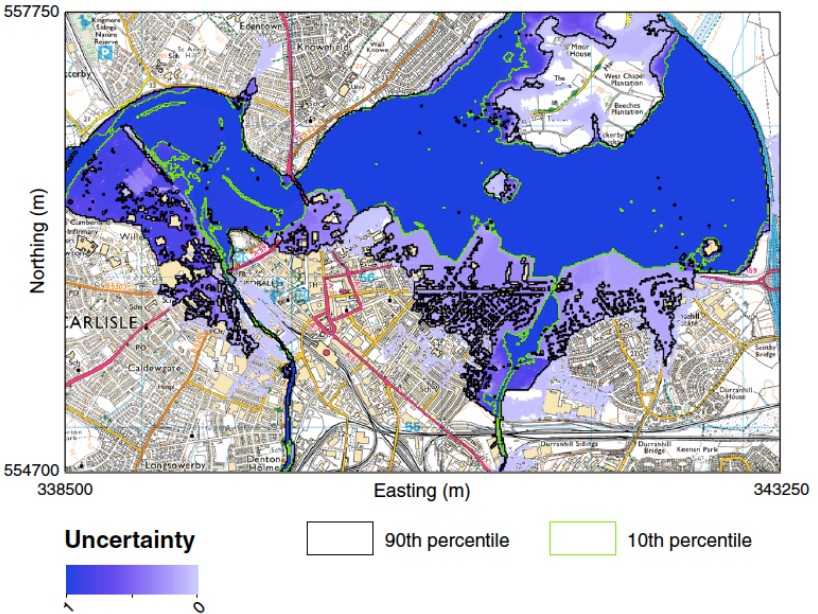

**Figure 2: Uncertainty in inundation extent  resulting from simulations of the flood with annual exceedance probability 0.01, River Eden valley in the vicinity of Carlisle, Cumbria, UK.  The uncertainty scale results from a behavioural ensemble of LISFLOOD-**



FP inundation models with different parameters sets, weighted according to fit to the 2005 flood outline, and driven by realisations from the joint distribution of peak discharges in the River Eden and the Caldew and Petteril tributaries (for full details see Neal et al., 2013).

In many situations, flooding is constrained by the existence of natural levees or artificial flood defences. Such defences are always associated with a residual risk of being overtopped and/or failing, a risk that will vary with the construction methods, programme of maintenance, unauthorised modifications and other factors (van Gelder and Vrijling, 2014). These are all subject to epistemic uncertainties, but are often dealt with through using fragility curves that give a probability of failure as a function of water level. Although expressed in terms of probabilities, such fragility curves are often treated as

deterministically known. The difficulties of including epistemic uncertainties are discussed, for example, by Gouldby et al. (2010) who used forward uncertainty estimation to cascade uncertainty assumptions through the RASP framework used in the U.K. National Flood Risk Assessments (NaFRA).

### 2.3 Uncertainty quantification in real-time flood management

In flood incident management, epistemic uncertainties might lead to deterministic predictions being quite biased, even where models of flood discharges and extent of inundation have been calibrated for past events. This is usually handled in one of two ways. Traditionally it was handled by the experience and expertise of the flood forecasters who would make subjective adjustments to model outputs available to them as an event progressed and more information became available. In doing so they would qualitatively allow for perceived epistemic uncertainties based on past experience. This approach is still used in

many countries. An extension of this approach is to base estimates of the uncertainty in model predictions based on the performance of the model in past events. A method such as quantile regression can be used for this (Lopez Lopez et al., 2014). The problem for both approaches is that past experience may not be a good guide to the peculiarities of a new event.

An alternative approach is to assume that all uncertainties can be treated statistically and use a data assimilation approach to

correct for over or under-prediction as the event proceeds. Techniques such as the Kalman filter, or stochastic autoregressive modelling, can be used with the advantage that an estimate of the variance of the forecast can also be updated at the same time (see for example, Sene et al., 2014; Young et al., 2014; Smith et al., 2012; 2013a). No explicit account of potential epistemic uncertainties is made in this approach; the aim is only to improve the forecast and minimize the forecast variance at the required lead-time as new data become available. The approach will often work well when the

required lead-time is less than the response time of the upstream catchment so that the data assimilation can rely on measured inputs. It works less well in flash flood situations in small catchments with short response times so that forecasts of the inputs are needed to produce a forecast with reasonable response time (Alfieri et al., 2011; Smith et al., 2013b). Rainfall forecasts from Numerical Weather Prediction (NWP) models are still not sufficiently accurate for this purpose but



are now used routinely (such as in the European Flood Awareness System hosted at ECMWF, Bartholmes et al., 2009; De Roo et al., 2011) for providing flood alerts some days ahead.

### 2.4 Flood Frequency and The Safety of Dams

The safety of dams is one example of a hazard that involves both natural forcing and engineering design, but one in which the consequences of failure can be catastrophic.   Failures can be due to poor engineering design, poor geological assessments of the location, poor maintenance of the structure, or an extreme flood event or landslide into the reservoir. Lists of dam failures (Vogel, 2001, see also http://www.damsafety.org/news/?p=412f29c8-3fd8-4529-b5c9-8d47364c1f3e,

http://en.wikipedia.org/wiki/Dam_failure) show that such events are not common, but the International Commission on Large Dams (ICOLD, 1995) has estimated that some 0.5% of all dams failed in the period 1951-1986 and there have been cases with hundreds or thousands of fatalities downstream.   There have, no doubt, been many other cases of near failure.   A recent example is in the Sheffield area of England where the Ully dam came close to failure as a result of erosion of the downstream dam face during a period of extreme rainfalls in 2007.   In the same area, the failure of the Dale Dike dam in

1864, while being filled for the first time, caused 244 fatalities and destroyed 600 houses (Smith et al., 2014).  The most fatalities estimated are for the failure of several dams in Henan Province in China in 1975 which killed an estimated 171,000 people and destroyed the houses of 11 million people, following prolonged heavy rain.   A well-known European example was the landslide-induced failure of the Malpasset arch dam in France in 1959 that caused the deaths of 423 people (Londe, 1987; Duffaut, 2014).

Multiple causes make dam failures difficult to predict, and most countries take a highly precautionary approach to regulating for dam safety.    The design of the dam and spillway channels for large dams are commonly designed to cope with the estimate of the flood with an annual exceedance probability of 0.0001.    This is a much smaller probability than for designing normal flood defences, because of the potential consequences of a failure.    Thus there is an epistemic issue of

defining of such an extreme event, and also what characteristics of such an event might impose the greatest forcing on the dam.    The greatest forcing might not come from the highest flood peak if it is of short duration, but from the inflow volume associated with an event of longer duration but smaller peak.    One way of assessing such effects is to run a continuous simulation model and examine the impact of the most extreme events generated over with long realisations (e.g. Blazkova and Beven, 2009).  The continuous simulation approach means that the antecedent conditions prior to any event are handled

naturally, but clearly the outputs from such simulations are dependent on the epistemic uncertainties associated with all the model components, including the tail assumptions for the driving distributions, the choice of rainfall-runoff model, and the estimation of model parameters given the historical data.    A particular feature of fitting such a stochastic model is that





whether a model appears to give a good fit to the observed statistics might depend on the particular realisation of generated inputs (Blazkova and Beven, 2009).

Predicting the downstream footprint of a dam failure and consequent potential damages can also be difficult.    There are
hydraulic models available designed to cope with the high discharges and sharp wave fronts expected with a dam failure (Cao et al., 2004; Xia et al., 2010), but the application in any real case study will depend on the epistemic uncertainty associated with the characteristics of a breach in the dam acting as an upstream boundary condition for the hydraulic model and the momentum losses in the downstream area as a highly sediment-laden fluid interacts with the valley bottom infrastructure and vegetation.    It is also difficult to verify the outputs of such a model (except for small scale physical
experiments in the laboratory, though see Hevouet and Petitjean, 1999; Begnudelli and Sanders, 2007; and Gallegos et al., 2009; for examples of field scale validation) while many damage assessment schemes are based on predictions of flood depths.   In the dam break case, velocities will also be important in the threat to life and damage to buildings.

## 3. Landslides and Debris Flows
### 3.1 Landslides and key epistemic uncertainties
Globally, landslides are directly responsible for several thousand deaths per year (Petley, 2012). A widely cited example is that of the Welsh village of Aberfan, where a flowslide from a colliery spoil tip killed 144 people, 116 of whom were children, at the Pantglas Junior School in October 1966 (Johnes, 2000). More recently, the Gunsu mudslide that occurred
after heavy rain in August 2010 in China, killed an estimated 1765 people. However, despite the large risks posed by landslides, the ability of research to guide and inform management decisions is limited by high levels of uncertainty in model assessments of slope stability. In landslide risk assessment epistemic uncertainties arise from a range of sources, including errors in measurement data, gaps in the understanding of landslide processes and their representation in models, and from uncertain projections of future socio-economic and biophysical conditions (Lee and Jones, 2004).

### 3.2 Uncertainty quantification in landslide hazard estimation
Landslide risk can be assessed qualitatively or quantitatively. The choice depends on the scale of work (national, regional, local or site-specific), and also on the quality and quantity of data available. For site-specific slopes, physically-based deterministic models centred on slope stability analysis are commonly used to assess the probability of landslide occurrence.
Stability conditions are generally evaluated by means of limit equilibrium methods, where the available soil strength and the destabilising effect of gravity are compared in order to calculate a measure of the relative stability of the slope known as the factor of safety. The limit equilibrium method relies on significant simplifications, such as that the failing soil mass is rigid, the failure surface is known and the material's failure criterion is verified for each point along this surface.   These simplifications limit both accuracy and applicability. Epistemic uncertainties related to the limited understanding of system





processes and functioning can lead to large errors in such model predictions. For example, in 1984 an embankment dam in Carsington, England, slipped, despite the fact that limit equilibrium analysis had indicated that the slope was not expected to be at risk of failure. This discrepancy has been shown to be caused by epistemic errors, as brittle soils may exhibit strain-softening behaviour when loaded, leading to progressive failure, a phenomenon which cannot be reproduced using
conventional limit equilibrium stability analyses. For this type of soil, finite element analysis using appropriate numerical algorithms and constitutive models are required to achieve a more accurate prediction of stability (Potts et al., 1990).

All physically-based slope stability models are subject to epistemic uncertainties in both the constitutive relationships chosen and the parameter values required by those relationships.   Parameter variability is often assessed by making small scale
laboratory measurements of parameters such as cohesion and coefficient of friction but the resulting values may not be directly applicable at the large scale because of the effects of spatial heterogeneities, and additional factors such as root strength (Christian et al., 1994; Rubio et al., 2004; Hall et al., 2004; Hürlimann et al., 2008; Hencher, 2010).   Although spatial variability of soil properties has been recognised as an important source of epistemic uncertainty in the literature (e.g. El-Ramly et al., 2002; Griffiths and Fenton, 2004), it has often been ignored in previous analyses using limit equilibrium
methods.  The use of constant values for soil properties over soil deposits may lead to unreliable estimates of the probability of failure of a slope (El-Ramly et al., 2002; Griffiths and Fenton, 2004; Cho, 2007; Griffiths et al., 2009). To account for this source of uncertainty in slope stability problems, some investigators combine limit equilibrium methods with random field theory (e.g. *Cho*, 2007). Random field theory allows soil properties to be described by a randomly generated distribution, instead of a single value across the entire modelled space. Moreover, random field theory also allows spatial correlation to be
preserved, ensuring that the values of a given property in adjacent slices do not differ as much as between slices, which are further apart.  It represents, however, a treatment of a source of epistemic uncertainty as if it was aleatory random variabity.

Given that limit equilibrium methods are based on a two-dimensional analysis where the critical failure surface is a line of arbitrary shape, the influence of the random field is only considered along that line and therefore, when this method of
analysis is used, it can be seen as a one-dimensional approach (Griffiths et al., 2009). To overcome this limitation inherent to limit equilibrium methods, while accounting for spatial variability of soil properties, Griffiths et al. (2009) suggest combining a finite-element model with random field theory. The finite-element method is particularly attractive as, in addition to satisfying equilibrium and compatibility, it allows any constitutive framework to be used for the simulation of the mechanical behaviour of the geomaterial.  The decision then is what constitutive framework to employ to allow more
accurate predictions to be obtained.

The finite-element method has the added advantage of being capable of simulating water flow and coupled hydro-mechanical behaviour under saturated and unsaturated conditions (Alonso et al., 2003; Gens, 2010). Time-varying boundary conditions to simulate the effect of rainfall and vegetation can be used (e.g. Nyambayo and Potts, 2010), although this will





require greater computational power and may still not represent the water flow processes adequately, such as the role of preferential flows and bedrock fracture systems in inducing conditions for failure (e.g Montgomery et al., 2009; Hencher, 2010; Beven, 2010; Beven and Germann, 2013). Even at sites where the costs of extensive field investigations can be justified, there is much that remains unknown about the subsurface including the detail of water flow pathways and
knowledge of the hydro-mechanical behaviour of soils.

To accommodate uncertainty caused by parameter variability in both limit equilibrium and finite-element methods of analysis, Monte Carlo simulation and/or the first-order-second-moment (FOSM) method are commonly used (e.g. Christian et al., 1994; Wu and Abdel-Latif, 2000; Haneberg, 2004; Cho, 2007). These methods consider the uncertainties introduced
by the inputs in different ways. Monte Carlo simulation starts by repeatedly sampling from the probability distributions of the random variables. A deterministic computation on each of generated input set is performed, and the factor of safety calculated. Subsequently, the aggregated results of all sets provide an approximation of the probability distribution of the factor of safety. Alternatively, the FOSM method can be used to estimate the probability of slope failure. This probabilistic method determines the stochastic moments of the performance function. As the input variables are randomly distributed, the
performance function is also randomly distributed, which the FOSM method characterises in terms of its mean and standard deviation. In both methods, therefore, the uncertain parameters are treated as aleatory variables.

Detailed slope stability models require geotechnical information on site conditions that can be prohibitively costly to obtain and so tend to be employed only in small areas for cases where high risk is anticipated. Over large and complex areas, where
the use of detailed physically-based models is not feasible, statistical/data-driven models relating the probability of spatial landslide occurrence (i.e. susceptibility) and local geo-environmental conditions (e.g. geological, topographical and land-cover conditions) are used instead (e.g. Guzzetti et al., 1999; Ercanoglu and Gokceoglu, 2002; Guzzetti et al., 2005, 2006). These models have become standard in landslide susceptibility assessment at a regional scale (Corominas et al., 2014). By estimating where the slope is most likely to fail (but not the recurrence of failure, i.e. the temporal frequency, or magnitude
of the expected landslide), these models can be of great help in land-use planning, guiding planners in the delimitation of suitable areas for future development.

Guzzetti et al. (2006), for example, established for the Collazzone area, Italy, a landslide susceptibility model through discriminant analysis by finding a combination of predictor variables that maximises the difference between the populations
of stable and unstable slopes with minimal error. The generalisation of a very complex problem into a relatively simple statistical model, necessarily introduces errors in model predictions, arising from errors in the predictors used to establish the model, uncertainty in the classification of the terrain units, etc. To estimate uncertainty, Guzzetti et al. (2006) suggest a bootstrapping re-sampling technique. Several landslide susceptibility models are determined, by varying the selected terrain units considered. These ensembles of models are run for the study area and descriptive statistics for estimated susceptibility,



including the mean and the standard deviation, are determined. A model relating the mean value and 2σ (a proxy for the model error) is fitted using a least squares method. This model is used subsequently to provide local estimates of the model error.

Another large source of uncertainty affecting the assessment of landslide susceptibility is often introduced by the imprecision with which experts approach a problem. To account for the uncertain and inexact character of the available information and for the possibility of limited information concerning a real system, fuzzy-based risk assessment models have been suggested in the literature (e.g. Ercanoglu and Gokceoglu, 2002; Lin et al., 2012). For example, based on a landslide inventory database, Ercanoglu and Gokceoglu (2002) applied factor analysis to determine the important weights of the factors

conditioning landslides in the area (slope angle, land use, topographical elevation, dip direction of movement, water conditions and weathering depth). Fuzzy-set theory is then applied, accounting for the judgemental uncertainty (fuzziness, vagueness, imprecision) introduced by the way experts approach the problem. In a rule-based fuzzy model, the fuzzy prepositions are represented by an implication function (e.g. 'If slope angle is very low then landslide susceptibility is non-susceptible') commonly called fuzzy if-then rules or fuzzy conditional statements. The fuzzy if-then rules are then used to

produce a fuzzified index map for each factor conditioning landslides. These maps are thereafter combined (by overlaying) to produce a landslide susceptibility map.

In the context of long-term landslide risk management, as for other natural hazards fields, such as floods or earthquakes, the probability of exceedance is often calculated for different sizes of events in terms of an extreme value distribution. This

approach has advantages over a simulation-based analysis, the results of which may be affected by uncertainties in input forcing data. However, this does not mean that uncertainties in factors contributing to landslides are ignored in probabilistic estimates of landslide risk. Instead, probabilistic estimates implicitly account for input uncertainty by fitting the statistical distribution of events to available historical data. The epistemic uncertainty is convolved into a question of what statistical distribution should be used and how uncertainty in the tail behaviour is estimated. Probabilistic models such as binomial

model, Poisson model (Crovelli, 2000) and the power-law distribution (Hungr et al., 1999; Dussauge-Peisser et al., 2002) have been suggested in the literature to estimate the frequency (or return period) of landslides of a given size.

### 3.3 Uncertainty quantification in real-time landslide warning systems

In the context of real-time warning systems, slope failure is commonly estimated by establishing landslide-triggering

thresholds of the initiating agent. The application of triggering thresholds has been used, for example, in early warning systems in areas prone to rainfall-induced landslides, by establishing relationships between landslide occurrence and rainfall indicators, such as antecedent rainfall, duration, intensity and cumulative rainfall (Aleotti, 2004; Cepeda et al., 2012). An empirical model between rainfall and landslide initiation has been used to issue warnings during the storms of 12 to 21 February 1986 in the San Francisco Bay Region (Keefer et al., 1987). Since information regarding data quality is often

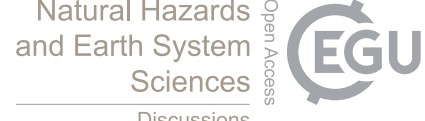

lacking, one common way to deal with uncertainty involves tracing the rainfall threshold curves that correspond to different percentiles and then deciding on a minimum threshold satisfying some performance criterion (e.g. rainfall threshold curve established so that includes 90% of the historical events) (Aleotti, 2004). Nevertheless, epistemic uncertainty introduced by lack of knowledge on landslide occurrence can be significant. Gariano et al. (2015) show that even a small (1%)

underestimation in the number of the considered landslides can result in a significant decrease in performance of an early warning system.

## 4. Droughts

### 4.1 Droughts and key epistemic uncertainties

Drought has the potential to cause widespread fatality and economic damage, particularly when a drought event might last for years or even decades. As with floods, droughts may be characterised either in terms of their natural severity or their impacts.  The definition of drought depends on the type of water deficit being considered (rainfall, stream flow etc.). Drought follows the hydrological cycle, as precipitation deficits (meteorological droughts) lead to low soil moisture levels (agricultural/soil drought) and decreased river flows (hydrological drought) which in turn may lead to lowering of reservoir

levels and water shortages (socioeconomic drought).   Drought periods associated with high temperatures may also have other impacts such as the large number of excess deaths in Europe in the summer of 2003 (Robine et al., 2003). Epistemic uncertainties in drought hazards stem from unknown future climate conditions, from unknown future water demand scenarios and lack of knowledge about how society might respond to long-term droughts, from low flow measurements with poorly understood errors, and from structural errors in hydrological models used to assess the impact of potential future

rainfall deficiencies. Epistemic uncertainties in estimates drought-related consequences and losses stem from the scarcity of data on and the difficult valuation of impacts and damage induced by water shortages.

### 4.2 Uncertainty quantification in drought hazard estimation

Drought hazard is widely assessed using indices, such as the standardised precipitation index (SPI) or Palmer Drought

Severity Index (PDSI). The most straightforward of these consider single environmental variables, such as precipitation (SPI) or groundwater level (Standardised Groundwater Index, Bloomfield et al 2013). In such cases sources of uncertainty are restricted to commensurability of recorded observation, which may arise for instance from missing data, incomplete or short records (Hong et al., 2014; Hu et al., 2014). However, the information content of such indices can be low as rainfall or groundwater levels are not the sole drivers of drought impacts.  By contrast, more complex indices such as PDSI and the

Crop Moisture Index provide a more applicable representation of drought, but with more sources of potential uncertainty due to multiple data sources, parameterizations, and model structures imposed by the indices. For instance, the Palmer Drought Severity Index or the Crop Moisture Index assume that land use and soil properties are uniform over large spatial scales; which makes it difficult to accurately identify the spatial extent affected by a drought (Narasimhan and Srinivasan, 2005). Parameter uncertainty in some drought indices is rarely considered when characterising drought, yet it has been shown to





play a significant role in the identification of major drought events and in the derivation of relevant drought statistics (Samaniego et al., 2013).

Under specific local conditions, shortage of rainfall can have an influence on water availability for human use at a regional
scale within 4 months (Marsh et al. 2007). Long droughts can be difficult to characterise as multiple periods of drought can be interrupted by wet weather events, without sufficient rainfall arriving to restore water storage. Acknowledging this, long drought events such as the 1890-1910 drought in England and Wales and the Millennium drought in Australia can be pernicious, gradually depleting water stored in aquifers and reservoirs. Historically, drought indices and other water availability metrics such as Deployable Output (DO) in the UK have been presented without associated quantification of
uncertainty. This is unfortunate, both in terms of the complexity of the calculation of such figures and as these terms are widely adopted by legal and regulatory systems. Recently, a risk-based approach has been proposed by Hall et al. (2012). Under this approach, probabilistic uncertainties are considered explicitly within the model and simulations are based on environmental time series, allowing metrics such as the probability of water shortages to be determined. This allows uncertainties to be examined simultaneously – conditional on the time series used to inform the model being representative
of those driving the real system.   As with other hazard areas, defining the probabilities required may also be subject to lack of knowledge.

Estimation of stream flow, and in particular low flows, is essential for hydrological drought analysis, thus the choice of methods to model and estimate low flow characteristics can introduce epistemic uncertainties in drought risk assessment.
Distributions fitted to low flows are susceptible to bias introduced by the fitting methodology and distribution choice (Ries and Friesz, 2000). Uncertainty is introduced in observations because many river gauging methodologies are especially poor at recording low flows (Barmah and Varley, 2012; Tomkins 2014; Coxon et al., 2015). As gauging methods record proxy observations of flow, epistemic uncertainty in functional relationships (i.e. changes in channel cross-section or vegetation affecting the correlation between stage and discharge) is likely to have a relatively greater effect on the absolute errors of
low flow observations (Tomkins 2014; McMillan and Westerberg, 2015). While there is significant attention paid to information-rich events such as recession rates following flood events, the assumption that recession parameters determined in this way are optimal for determining the hydrology of extended low flow series is not valid (Prudhomme et al. 2012, 2013). Hydrological models, which are routinely applied to model low flow occurrence and to characterise hydrological drought duration and deficits in response to particular climatological conditions, also introduce epistemic uncertainty in
drought risk assessments. For example, Duan and Mei (2014) have shown that hydrological model structural uncertainty induces large differences in drought simulation.

Drought risk can be characterised using metrics of drought duration and intensity (the deficit of water during a drought event), or the joint probability of a sequence of reduced flow events either in isolation or in combination with a water supply




system model to assess future drought risk. Drought duration is indicative of drought severity rather than directly responsible for consequence in itself, as a long period of low flow is not necessarily worse than a short, sharp drought. Intensity can be considered a more robust metric of shortage as deviation from a threshold state can develop as a consequence of brief periods of extreme shortfall, longer mild shortfall or some combination of the two. Both these methods are sensitive to the

identification of a threshold, which can be non-stationary due to environmental factors. Autocorrelation in drought series can be difficult to identify due to the requirement of capturing both the different temporal scales (daily, annual) and the continuous range of low flows, as correlation in Q99 events may be independent from correlation in Q95 events).

Epistemic uncertainties related to future climate conditions influence drought risk assessment for water resource planning
purposes. A number of studies have investigated forward uncertainty analysis of the potential impacts of climate change on droughts (e.g. Wilby and Harris, 2006). Borgomeo et al. (2014) developed a risk-based method to incorporate epistemic uncertainties related to climate change in water resources planning and to assess drought and water shortage risk in water supply systems. This risk-based method incorporates climate change epistemic uncertainty by sampling the United Kingdom Climate Projections (UKCP09) change factor distribution. Sampling different vectors of change factors allows for
exploration of some degree of epistemic uncertainty in future climate, within the range of the UKCP09 scenarios. Similarly, climate model information was used by Paton et al. (2013) to assess drought risk in the southern Adelaide water supply system, Australia. In that study some of the climate-related epistemic uncertainty was accounted for by developing hydro-climatological scenarios based on six different greenhouse gas emissions trajectories and several general circulation models.

Although climate models may provide information about future drought risks, there are issues here about how far current climate models can reproduce the type of blocking high-pressure conditions that lead to significant droughts in Europe. In addition, the probabilities of multi-year droughts under future climates will almost certainly be poorly estimated. In this context, the historical periods of 1933-1934 and 1975-1976 in the UK are still used as extreme cases for water resource planning purposes. This is a form of precautionary approach, that does not require any estimate of probability associated
with that event, but one which involves some epistemic uncertainty about whether a more extreme event might occur in future. Similar worst-case scenario approaches have been applied by Kasprzyk et al. (2009) and Harou et al. (2010) to assess drought risk and evaluate drought management strategies in water resources supply systems undergoing change when, in addition to any climate changes, human interventions modify exposure, vulnerability etc. (i.e., the non-hazard related component of the risk equation) (Mechler et al., 2010).

## 5. Earthquakes
### 5.1 Earthquakes and key epistemic uncertainties
Predicting earthquake occurrence is difficult, especially large seismic events in the very near future. Recently, the 2011 Tōhoku earthquake in Japan has highlighted that estimation of the maximum magnitude of mega-thrust subduction



earthquakes involves significant epistemic uncertainty related to segmentation of seismic sources, which can lead to the gross underestimation of earthquake scenarios. On the other hand, during the 2010-2011 Christchurch sequences in New Zealand, the complex behaviour of interacting fault systems has caused clustering of multiple major events in the Canterbury region, also resulting in major economic impact. Generally, earthquake hazards are influenced by stochastic nature of

earthquake occurrence and their size as well as by uncertainties in ground motions at sites of interest, which are contributed by uncertainties in source, path and site characteristics. A standard approach for characterising potential future earthquakes is Probabilistic Seismic Hazard Analysis (PSHA; Cornell, 1968; McGuire, 2001, 2004). In PSHA, key uncertainties related to earthquake occurrence in time and space, earthquake magnitude, and ground motion prediction, are all captured. However, in the past, major earthquakes have often been surprises, indicating that our knowledge is not perfect; we learn

new things from these events and are sometimes urged to revise theories in the light of new observations.

**5.2 Uncertainty quantification in earthquake hazard estimation**

PSHA takes into account numerous earthquake sources and scenarios and integrates their contributions probabilistically as if all variables considered are aleatory in nature. The primary objective of PSHA is to develop a set of seismic hazard estimates

for aiding the revision and implementation of seismic design in national building codes. Outputs from PSHA are provided in various forms, such as site-specific hazard curves for safety-critical facilities and regional hazard contour map. The contour map, in contrast, shows expected ground motions (e.g. peak ground acceleration and spectral accelerations) across a wide area or region at a selected annual exceedance probability level (typically 1/500 to 1/10,000).

PSHA involves various types and sources of uncertainties, and thus it is crucial to adopt an adequate mathematical framework to handle uncertainties as probabilities for individual model components and their dependency (Woo, 2011). Physically, these uncertainties can be associated with earthquake occurrence processes in time and space, seismic wave propagation, and seismic effects on structures and socioeconomic systems. PSHA also allows the identification of critical hazard scenarios at different probability levels through seismic disaggregation (McGuire, 2004). This essentially closes the

loop between probabilistic and deterministic seismic hazard approaches, which are complementary in nature (McGuire, 2001). The deterministic scenario approaches (e.g. Zuccolo et al., 2011) allow the use of more definitive models and data, but without attempting to associate a probability with a given scenario. For evaluating seismic risk impact to safety-critical facilities and infrastructure, both approaches should be implemented and should also be accompanied by rigorous sensitivity analysis.

Epistemic uncertainties arise both in the choice of structure for the component models and in the effective values of the parameters necessary. As with the other natural hazards, this means that when model predictions are compared to observational data the prediction errors can have a complex structure that may not be simply aleatory In PSHA, representations of alternative hypotheses and assumptions for individual model components are often framed with a logic





tree approach (Kulkarni et al., 1984), and the final estimates of seismic hazard parameters are obtained by integrating relevant uncertain model components and by probability weighting of alternative assumptions. A benefit of using a logic tree, despite its simplicity, is the transparency in characterising epistemic uncertainties. In this regard, the logic tree approach is similar to the condition tree of analysis assumptions outlined by Beven and Alcock (2012). Nevertheless, difficulties arise
because not all models, which analysts wish to apply, are based on consistent data or assumptions, and the probabilities of alternatives in the logic tree are often poorly known, unknown, or unknowable (Bommer, 2012; Stein and Stein, 2013).

Thus, in practice, given these epistemic sources of uncertainty, it is not a trivial task to assign weights to individual branches of the constructed logic tree and, often, resorting to expert elicitation is the only solution. For major industrial facilities (e.g.
dams and nuclear power plants), the development of the logic tree is often carried out according to the Senior Seismic Hazard Analysis Committee (SSHAC) guidelines for using expert advice (Budnitz et al., 1997). In the face of epistemic uncertainties and wide spreads in experts' opinions, special care is essential to avoid the inflation of elicited uncertainties and parameter distributions (Aspinall and Cooke, 2013).

Two of the critical elements in PSHA, which are linked but both subject to considerable epistemic uncertainties, are the estimation of long-term occurrence rates of large earthquakes and the evaluation of the maximum magnitude for use in a PSHA, for a given seismotectonic environment. On occasion, the upper bound of the maximum magnitude may not be constrained either physically or statistically (Kagan and Jackson, 2013). The difficulty simply stems from the fact that records of seismicity data are insufficient to derive such long-term occurrence rates reliably, solely from historical
catalogues or instrumental databases. The quality, completeness and reliability of an earthquake catalogue evolves over time, affected by the distribution of human settlements and the way in which major events in the historical record have been reported, and by advances in measurement technology and, more recently, the wider geographical coverage of seismographic networks. This often results in inhomogeneous detection and monitoring capabilities of instrumental catalogues (Tiampo et al., 2007), which needs to be accounted for in evaluating earthquake occurrence rates. In addition, new information from
terrestrial and ocean geodesy (McCaffrey et al., 2013; Bürgmann and Chadwell, 2014) will help constrain seismic hazard estimates derived from PSHA.

Estimating frequency of occurrence of events for an individual fault or fault system and their magnitudes is highly uncertain and depends strongly on assumptions (Murray and Segall, 2002). In particular, it is difficult to determine the continuity of
fault segmentation (Shen et al., 2009). In such cases, different hypotheses regarding the rupture behaviour of the fault system may be represented by branches of a logic tree. Recent PSHA studies for potentially active but less well-instrumented seismically active regions (e.g. the East African Rift) have extended the modelling basis for regional seismicity beyond historical and instrumental earthquake catalogues by using information from mapped geological faults and geodetically-determined rates of strain accumulation (e.g. Hodge et al., 2015). It is noteworthy that while such PSHA assessments remain




significantly uncertain, they may be better able to capture potential extreme (surprise) events. Rigorous sensitivity analysis should include testing alternative hypotheses and comparing the impacts of the adopted assumptions on regional seismic hazard assessments (see, for example, the flooding example by Savage et al., 2016). In this regard, a PSHA should be reviewed, even from a modern instrumental perspective, such that a better understanding of seismic hazard assessments and

their uncertainties can be achieved (Woo and Aspinall, 2015).

It has become more established in recent years that the mean occurrence rates of earthquakes on many mature fault systems and in subduction zones (where multiple plates meet and interact) are non-Poissonian and quasi-periodic (in contrast with a homogeneous Poisson model in the classical formulation of PSHA), and thus the hazard and risk potential posed by specific

faults or subduction zones may be regarded as time-dependent (Sykes and Menke, 2006). Both physics-driven occurrence models (Shimazaki and Nakata, 1980) and statistics-based renewal models (Cornell and Winterstein, 1988; Matthews et al., 2002) have been adopted in PSHA. A notable example of an active seismic region that is affected by a renewal earthquake process is the Cascadia subduction zone. A unique aspect of this subduction zone is that repeated occurrences of $M_w$9-class mega-thrust earthquakes - due to subduction plate motions - have been recognised from field evidence only relatively

recently (Satake et al., 2003; Goldfinger et al., 2012). In other words, the occurrence and rupture processes of the Cascadia subduction zone involve major epistemic uncertainties, and yet detailed hazard and risk assessments are necessary from an earthquake disaster preparedness viewpoint. In the last decade, various seismic hazard and risk studies for possible risk mitigation have been carried out by adopting a wide range of time-dependent models and possible rupture scenarios as a way of trying to account for sources of epistemic uncertainty (Goda and Hong, 2006; AIR Worldwide, 2013). This situation

contrasts with the case for the 2011 Tōhoku earthquake, where the consideration of extreme events was not taken up in risk mitigation actions prior to this event, even though there were indications of the impacts of past major tsunami-inducing events in the region. In this case and that of the Cascadia zone, current knowledge and understanding of subduction events are likely to be further updated in the very near future by seafloor geodesy, in particular, and so the scientific assessment framework and tools for better quantifying the characteristics and patterns of such earthquakes should also evolve

dynamically.

Characterising seismicity for the purposes of PSHA is always challenging, even in areas with plentiful data, and even more so when it comes to estimating background or diffuse seismicity away from known active regions or in low seismicity areas. Conventionally, this has been tackled, following Cornell (1968), by developing an area source zone model, each component

of which is associated with an annual occurrence rate (above a minimum magnitude) and a Gutenberg-Richter type magnitude distribution. However, because earthquakes are a manifestation of a geological process, epistemic uncertainties in relation to earthquake magnitude-occurrence rates – especially at high magnitudes – should not be derived solely from the statistical properties of recent monitoring datasets or even historical catalogue information, either of which is just a limited snapshot sample of the underlying process. The danger here is that the analyst, in considering how to characterise a

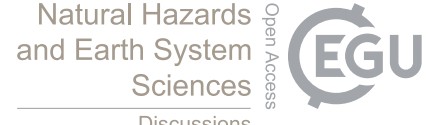



seismicity model for PSHA, is seduced into deriving a model conditioned on the available data, rather than understanding the probative weight of that data given an infinitude of plausible causal process models: naively letting "the data speak for itself" in PSHA can easily be undermined by future events, as evinced by the Tōhoku earthquake. Thus epistemic uncertainty quantification of seismicity should be based on a wider assessment that integrates in other, difficult aspects, using expert

judgment - such as slip and strain/stress rates and geological and tectonic controls - in order to supplement the limitations of available data (Aspinall, 2013; Aspinall and Cooke, 2013). This precept applies equally, or should do, to other factors and parameters in a PSHA, e.g. maximum magnitude and focal depth distribution. The corollary to this, in practice, is that rigorous sensitivity testing of input parameters can provide a wider perspective for epistemic uncertainty in earthquake occurrence characterisation.

In modern practice, considerable effort has been invested in respect of ground motion prediction equations, which constitute another major source of uncertainties in PSHA. Empirically derived prediction models using observed strong motion records are inherently limited by the availability of such data. Even following the dramatic expansions of strong motion networks in active seismic regions (e.g. California and Japan), near-source strong motion data and strong motion data for very large

earthquakes (with the notable exception of the 2011 Tōhoku earthquake) are still lacking.  This reality forces us to update existing empirical ground motion models from time-to-time by incorporating newly available data or to use computational model simulations of strong motion (e.g. Skarlatoudis et al., 2015).  Another important issue, related to ground motion modelling using observed records, is that the majority of the existing ground motion models have been developed based on the ergodic assumption (Anderson and Brune, 1999).  The ergodic assumption in the context of ground motion modelling

implies that the ground motions required at a specific location can be substituted by recorded ground motions at different locations. There may be limited physical validity for this assumption in reality and, at best, adopting it *faute de mieux* engenders exaggerated epistemic uncertainty in the site-specific case via regression scatter estimates. In practice, the consequences of adopting this working hypothesis are biased seismic hazard assessments (Atkinson, 2006).

A new generation of ground motion models addresses the problem more rigorously (Stafford, 2014).  New strong motion data also offer new insights regarding the earthquake source processes via source inversion (e.g. slip distribution and asperities, i.e. concentrated slip patches; Mai and Beroza, 2002; Lavallee et al., 2006).  The improved knowledge of the earthquake source process in turn necessitates updated definitions of the seismological parameters and their use in PSHA. For instance, the asperity-based earthquake source model requires additional parameters to characterise the location and

concentration of earthquake slips within a fault plane. Furthermore, measures that are used to represent the source-to-site distance (e.g. hypocentral distance and rupture distance) may not be relevant within the extended methodology and may introduce epistemic uncertainties in ground motion predictions for future earthquakes (Goda and Atkinson, 2014). New data and theories facilitate the refinements of existing ground models and PSHA methods in a complex manner that may require

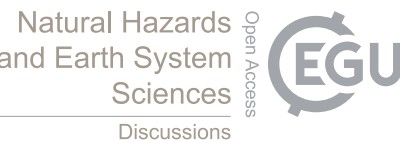

the specification of additional uncertain parameters. Further uncertainties may be introduced by the nonlinear relationships between ground motion characteristics and structural response and damage in assessing seismic risk.

In summary, while seismic hazard assessment is beset by some very challenging epistemic uncertainties, it is an area where
perhaps more progress has been made than for most other natural hazards, especially in respect of extreme and low probability events within the assumptions of a probabilistic analysis informed by expert elicitation.

## 6. Tsunamis

### 6.1 Tsunamis and key epistemic uncertainties

Massive tsunamis triggered by large earthquakes pose major threats to modern society, generating fatalities, disrupting socioeconomic activities, and causing grave economic impact across the world. Forecasting tsunamigenic earthquakes is challenging for the same reasons discussed above for prediction of mega-thrust earthquakes. Major sources of epistemic uncertainties are related to earthquake rupture processes (e.g. source areas and size, asperity, and kinematic/dynamic rupture process) and inundation/run-up process (e.g. topographical effects, land surface friction, and flow dynamics in urban areas).

### 6.2 Uncertainty quantification in tsunami hazard estimation

Estimating potential earthquake size is one of the most critical factors in predicting the impact of great tsunamis. Inappropriate application of seismological theories could result in gross underestimation of earthquake magnitude of mega-thrust subduction earthquakes (Kagan and Jackson, 2013). Moreover, the earthquake rupture process is complex and highly
uncertain, and is governed by pre-rupture stress conditions and frictional properties of the fault that are largely unknown/unobservable and heterogeneous in space. A large earthquake may also trigger a submarine landslide, which acts as secondary sources for tsunami generation (Tappin et al., 2014). To gain further insights into the earthquake rupture process, source inversions can be carried out to characterise the space-time evolution of the rupture by matching key features of simulated data with observations. Although sophisticated mathematical frameworks for source inversion have been
developed and implemented, derived earthquake rupture models vary significantly, depending on the methods and data used for inversion (Mai and Beroza, 2002; Lavallee et al., 2006). For instance, location, size, shape, and amplitude of slip asperities differ significantly among inversion models for the 2011 Tōhoku earthquake (Goda et al., 2014), reflecting the complexity and uncertainty in imaging the rupture process for mega-thrust subduction earthquakes. Additionally, different source modelling approaches, such as surface rupture to ocean bottom, effects of horizontal deformation of steep slopes on
vertical deformation, hydrodynamic response of water column, and time-dependent rupture process, slow versus fast rupture propagation speed, will influence the resulting tsunami waves (Geist, 2002; McCloskey et al., 2008; Løvholt et al., 2012; Satake et al., 2013). All these factors contribute to epistemic uncertainties related to tsunami source modelling.





Topographical features of near- and on-shore areas have major effects on tsunami waves and inundation/run-up. The spatial resolution and accuracy of bathymetry and digital elevation models (DEM) are important for representing local terrain features realistically. Typically, the frictional properties of terrain features are modelled by Manning's roughness coefficients. Different data resolutions will require different effective roughness coefficients, thus affecting tsunami

inundation extents. The impacts of uncertainty in the DEM and roughness coefficients will depend on tsunami hazard parameters (Kaiser et al., 2011). For instance, the inundation depths are less sensitive to the data resolutions and characteristics, whereas the flow velocity and momentum, which are also important in evaluating the tsunami-induced forces on buildings (Koshimura et al., 2009), are more sensitive. This issue becomes even more critical when tsunami inundation in dense urban areas is investigated, where buildings may be represented as (impermeable) elevation data. The simulated flow

velocities in urban streets can be very high.

It is rare that that uncertainties of the DEM data and roughness coefficients are taken into account in conducting tsunami simulations but adopting the same modelling philosophy as the PSHA of the last section, probabilistic tsunami hazard analysis (PTHA) has been developed and applied to some major tsunami-prone regions (e.g. Annaka et al., 2007; Thio et al.,

2007; Horspool et al., 2014). The main focus and advantage of PTHA are to integrate potential tsunami hazards from various sources (both near-field and far-field) in a probabilistic framework. Epistemic uncertainties are mathematically represented in PTHA through a logic-tree approach by assigning weights to alternatives for different model components. The final output is a tsunami hazard curve and probabilistic tsunami inundation maps of inundation depth and other relevant parameters. A major difference between PTHA and PSHA is that differential equations of tsunami wave propagation (typically shallow

water equations) in the ocean are evaluated directly, whereas in PSHA, seismic wave propagation (as well as earthquake rupture and site response) is approximated using empirical ground motion models. The direct simulation of tsunami waves reduces the uncertainties associated with tsunami hazard assessment, and provides additional information on the tsunami wave time-history and arrival time.

However, PTHA can be computationally demanding. To achieve computational efficiency, PTHA is often formulated based on linear superposition of tsunami waves (i.e. Green's functions) for simplified earthquake sources and is carried out only for near-shore locations (e.g. at 30 m depth). The inundation and run-up processes are often modelled by applying amplification factors (e.g. Løvholt et al., 2014). To improve the tsunami hazard prediction and quantify the effects of epistemic uncertainties, it is desirable to integrate the stochastic source modelling approach (which carries out fully nonlinear

inundation simulation of tsunami waves; Goda et al., 2014) into the PTHA methodology. De Risi and Goda (2016) have developed probabilistic earthquake-tsunami multi-hazard analysis based on the stochastic source modelling approach. Such an extended PTHA can reflect the variability of source characteristics for specific scenarios as well as numerous tsunami sources in developing tsunami hazard curves and maps.

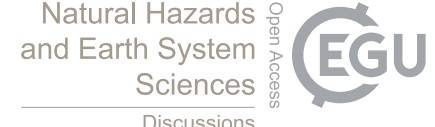



## 7. Volcanic eruptions and ash clouds

### 7.1 Volcanic eruptions, ash clouds and key epistemic uncertainties

The 2010 eruption of Eyjafjallajökull in Iceland provided a dramatic demonstration of the potential for volcanic ash clouds as a natural hazard. Because of the synoptic weather at the time of the eruption the ash cloud caused enormous disruption to
air travel across Europe and the Atlantic with some 10 million air travellers being affected. The total global cost in GDP over the entire eruption was estimated at $5 billion (Mazzocchi et al., 2010). Since then there has been considerable effort expended in the monitoring and prediction of volcanic ash clouds. Ash clouds can also be a problem in many other parts of the world, for example as a result of the continuing eruption of Mount Sinabung in Indonesia and the recent 2015 eruption of the Calbuco volcano in Chile. Globally, a network of nine Volcanic Ash Advisory Centres (VAACs) provides warning
services based on monitoring and modelling. Major epistemic uncertainties include the magnitude of the source term or mass emission rate, near field processes affecting ash size distribution and deposition, and the interaction of the eruption and synoptic weather patterns in affecting the far-field ash dispersion.

### 7.2 Uncertainty quantification in volcanic and ash cloud hazard estimation

Infrared satellite observations are perhaps the most important tools for monitoring ash but are not without their problems. Ash detection is complicated by a number of factors. The brightness temperature difference (BTD; the difference between brightness temperatures at two infrared channels) used as the basis for infrared ash detection can be affected by false positives and false negatives due to atmospheric conditions (Simpson et al., 2000; Mackie and Watson, 2015), land surface type and temperature, presence of other aerosols (Prata, 1989; Prata et al., 2001; Pavolonis et al., 2006; Lee et al., 2014) and
water/ice (Rose et al., 1995), in addition to particle size (e.g. Millington et al., 2012) and ash cloud opacity (Rose et al., 2001) (see Figure 3). Sophisticated volcanic ash retrieval schemes such as Francis et al. (2012) and Pavolonis et al. (2013) use a third infrared channel to help with removing false detections.

Many assumptions are made about the physical properties of ash in order to make estimates of other physical properties such
as ash column loading, ash cloud height and effective radius. For example, in the Met Office 1D-Variational (1D-Var) volcanic ash retrieval scheme (Francis et al., 2012) it is assumed that ash particles are spherical to simplify the absorption and scattering calculations, the particle size distribution (PSD) is assumed to be lognormal in shape, and the geometric standard deviation of the distribution is selected from a number of possible values. However, this value can have a significant effect on retrieved ash column loading e.g. (Western et al., 2015). Ash composition, and hence, refractive index
data must also be assumed, adding considerable uncertainty (Mackie et al., 2014). There are limited ash refractive index data available, and this choice can also have a significant effect on derived ash properties (e.g. Francis et al., 2012). The PSD geometric standard deviation and refractive index data set are varied within the 1D-Var algorithm and the solution with the lowest cost is generally used; the solution cost of the 1D-Var scheme can be used as an uncertainty measure, with high costs indicating high uncertainty (Stevenson et al., 2015).

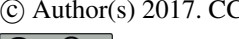


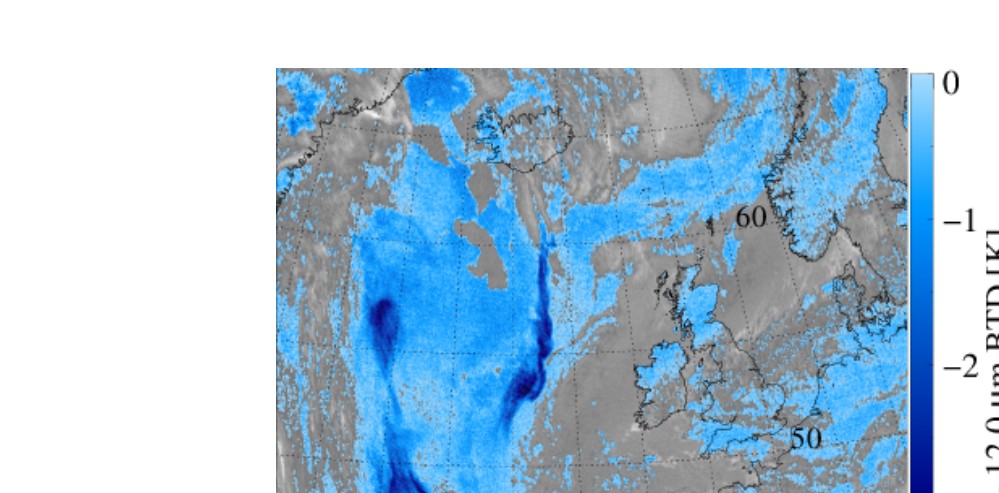

**Figure 3. Meteosat Second Generation Spinning Enhanced Visible and InfraRed Imager (SEVIRI) brightness temperature difference image (brightness temperature at the 10.8 μm channel minus the brightness temperature at the 12 μm channel) indicating the extent of the Eyjafjallajökull ash cloud at 0300 UTC 8th May 2010. The negative values of BTD (indicating ash) are shown in blue and the scale in Kelvin is given on the legend. The positive BTD is plotted in grey. A likely false negative ash signal can be seen south of Iceland where the ash plume appears to be obscured, possibly by meteorological cloud, due to the high ash concentration causing opaqueness, a large fraction of large particles or the presence of water in the plume. A negative BTD signal can be seen over North Africa / south eastern Spain, possibly due to a night-time clear arid land surface. Raw data supplied by EUMETSAT.**



Within volcanic ash retrieval schemes, other sources of uncertainty are introduced, for example in the simulation of satellite imagery using a radiative transfer model, in the meteorological data used within the model, interpolation of that data and so on. Some of these uncertainties are very generally accounted for the in 1D-Var algorithm but not all. Other types of observations (e.g. hyperspectral satellite observations, satellite, aircraft or ground-based lidar) can add useful information on

ash layer depth, particle size distribution and the height of the ash cloud. Combining these observations with infrared satellite observations can help reduce epistemic uncertainty in the derived observational data. However, they can often be of much lower temporal or spatial resolution and carry their own assumptions and uncertainties. Many of these sources of uncertainty are epistemic in nature and not necessarily aleatory in their characteristics.

Modelling of the hazard using volcanic ash dispersion models is, however, a problem of forecasting. At the UK Met Office the Numerical Atmospheric-dispersion Modelling Environment (NAME) model (Jones et al., 2007) is used in both simulation and forecasting of ash to inform the London VAAC which covers eruptions in Iceland and the impacts on northwest Europe. As with all models, NAME is a simplified representation of the problem, and does not include some of the complex physical processes that control the behaviour of an ash field close to the source of the eruption, notably fall out

of very large grains and particle aggregation. Near-field processes are still the subject of current research (e.g. Taddeucci et al., 2011). Currently, the effects of gravity currents (Bursik et al., 1992a; Sparks, 1986) are also not included in most atmospheric dispersion models. These near-source processes are likely to dominate ash dispersion and transport close to the source, and for large eruptions they could dominate for hundreds of kilometres (Bursik et al., 1992a, 1992b; Sparks et al., 1997), but far from the source are unlikely to affect downwind ash clouds for weak eruptions (Costa et al., 2013; Devenish et

al., 2012b).

In NAME an effective source term is used as a boundary condition for forecasting the far-field transport and deposition of ash. This includes assumptions about the PSD of the ash. Plume behaviour can vary significantly over time and information derived from deposited ash, often after an event, does not necessarily give a good indication of the PSD within the distal ash

cloud (Bonadonna and Houghton, 2005). Operationally, a default source term PSD has been used by the London VAAC, based on empirical measurements from Hobbs et al. (1991) which aims to represent the fine ash that survives near-source fall-out (Webster et al., 2012). This component may be of the order of 0.05 – 10 % of the total erupted mass (Mastin et al., 2009) and consequently constitutes a significant source of uncertainty. Mass emission rate (MER) and particle density are also required and are also very difficult to determine experimentally. MER is often represented as a simple empirical power

law as a function of plume height with fixed parameters (e.g. Mastin et al., 2009), while in a study of the Eyjafjallajökull eruption, Webster et al. (2012) used a fixed ash density value of 2300 kg m$^{-3}$. It is thought that the empirical function for MER may be biased towards observed data from larger eruptions (Woodhouse et al., 2013). Plume height measurements used to determine MER (e.g. radar) are subject to uncertainties (Arason et al., 2011; Folch et al., 2012), and plumes from weak eruptions such as Eyjafjallajökull can become distorted by local winds, increasing plume height measurement





uncertainty and therefore affecting the MER calculation (Webster et al., 2012). Meteorological data can also introduce uncertainty to dispersion forecasts, and can lead to cumulative transport errors (Dacre et al., 2016). All of these factors represent primary epistemic uncertainties in the application of such models. Even a cursory treatment of those uncertainties results in a significant predictive uncertainty (Devenish et al., 2012a).

One way of constraining such uncertainty is to use inversion modelling to learn more about model eruption source parameters (ESPs) (and possibly dispersion processes such as sedimentation, wet and dry deposition and atmospheric turbulence parameters), based on the available observations and prior information (e.g. Kristiansen et al., 2012; Moxnes et al., 2014; Pelley et al., 2015; Stohl et al., 2011). In this way, Kristiansen et al. (2012) estimated optimal volcanic ash source

terms for the Eyjafjallajökull eruption using an inversion algorithm with satellite-retrieved ash column loadings, a number of emission scenarios and two atmospheric dispersion models. The inversion-estimated source terms were applied within the models a posteriori to perform long-range forecasts and results were validated using LIDAR and in-situ PSD measurements from research flights. Uncertainties in the a priori emission estimates, model and observations were taken into account within the inversion algorithm, allowing the result to deviate from the a priori emission assumptions and the observations

according to the errors.

Wilkins et al. (2014, 2016) used data insertion to initialise NAME using measurement-derived data. Instead of releasing ash with a defined release rate from the volcano vent, it was released several times from "snapshots" of downwind ash clouds defined using retrieved data from infrared satellite imagery, in situ and other remotely sensed data. While this method does

not explicitly deal with uncertainties in the model or observations, it could potentially be used to bypass a lack of knowledge of the ESPs, for instance where the location of the volcano is unknown. The method does, however, require estimations of ash layer thickness, vertical distribution and PSD.

An inversion modelling based Bayesian method was adopted by Denlinger et al. (2012) to propagate uncertainty in ESPs

within an atmospheric dispersion model and estimate forecast uncertainty. A model was run with ESPs sampled from probability distributions. The model outputs were then assessed by comparison with satellite observations, a likelihood function defined, and a posterior probability distribution determined using Bayes theorem.  A genetic algorithm variational method was applied by Schmehl et al. (2011) to elucidate wind direction, wind speed and mass emission rate to be used for forward assimilation in a dispersion model. In this approach a cost function based on the difference between observed and

modelled fields was minimised over a series of iterations or until the solution converged. By sampling the source term parameter ranges iteratively, the results could be used to constrain uncertainty in ESPs and/or meteorological fields. Stefanescu et al. (2014) used ensembles of meteorological wind fields and ESPs to create weighted probabilistic forecasts of ash dispersion, which included contribution from a priori information and uncertainty in the wind and eruption source term. The ESP inputs were chosen so that the weighted forecast would give estimates of probability and variance in the model



output that would normally require application of statistics to a much larger number of randomly sampled inputs. So, the order of thousands of model runs was required instead of hundreds of thousands to millions, as would be required in a Monte Carlo method, for example.

.

**7.3 Uncertainty quantification in real-time volcanic and ash cloud hazard warning systems**

When such models are used for forecasting it is possible to compensate for epistemic uncertainties, at least can be in part, by the assimilation of information about the ash cloud derived from remote sensing and other direct sources such as experimental flights. Data assimilation will then implicitly compensate for some of the epistemic uncertainty associated with the model. However, the propagation of complex epistemic uncertainty in computationally expensive atmospheric-dispersion

models is a time consuming and difficult problem to quantify. The characterisation of volcanic ash forecast uncertainties in an operational time scale therefore remains a challenging task.

**8. Pyroclastic density currents**

**8.1 Pyroclastic density currents and key epistemic uncertainties**

Rapidly-moving flows of hot, fragmented gas-rich magmatic products in Pyroclastic Density Currents (PDC; also known as nuées ardentes or pyroclastic flows and surges) are the biggest killers in explosive volcanic eruptions. The 79CE eruption of Vesuvius and the remains found at Herculaneum and Pompeii represent a classic historic example of the disastrous impacts of PDCs, and any repeat at this volcano in the future, even on a smaller, less intense scale, could have massive consequences

for the heavily-populated surrounding area. Hazard and risk assessments for this situation, undertaken in the last twenty years for the National Emergency Plan (DPC, 1995, 2001), were mostly based on the characterisation of a single "Maximum Expected Event" (MEE). Such an event largely corresponds in the expected intensity of effects to the hazardous phenomena that occurred during the last sub-Plinian eruption of Vesuvius, in 1631CE. However, that definition was not based on a fully quantitative analysis of the whole system and potential ranges of eruptive activity, and no probabilistic estimates were

provided for the occurrence of the hazard events being considered. Much is still to be learned about the factors that control their initial formation, their movement across terrain and the ways they injure and kill people, and damage structures. Thus there are multiple sources of epistemic uncertainty about the hazards and risks associated with these dangerous phenomena.

**8.2 Uncertainty quantification in pyroclastic density currents hazard estimation**

In work for the EXPLORIS project (Neri et al., 2008), probabilistic characterisations of possible future eruptive scenarios at Vesuvius volcano were elaborated and organised within a risk-based framework, and a wide variety of topics relating to this basic problem were pursued: updates of historical data, reinterpretation of previous geological field data and the collection of new fieldwork results, the development of novel numerical modelling codes and of risk assessment techniques. To achieve coherence, many diverse strands of evidence had to be unified within a formalised structure, and linked together by





expert knowledge. For this purpose, a Vesuvius 'Event Tree' was created to summarise in a numerical-graphical form, at different levels of detail, all the relative likelihoods relating to the genesis and style of eruption, development and nature of volcanic hazards, and the probabilities of occurrence of different volcanic risks in the next eruption crisis. In order to achieve a complete parameterisation for this all-inclusive approach, exhaustive hazard and risk models were needed,

quantified with comprehensive uncertainty distributions for all factors involved, rather than simple 'best-estimate' or nominal values. Thus, a structured expert elicitation procedure was implemented to complement more traditional data analysis and interpretative approaches, and to add a formalised approach to the generic incorporation of epistemic uncertainty in the assessment by way of the Event Tree formulation.

Here, we focus on the issues of epistemic uncertainty in relation to the physical characterisation of PDC potential during a Sub-Plinian column collapse eruption, and how the topography of the volcano influences hazard and risk mapping results. Basic effort in this regard, during EXPLORIS, was dedicated to the development and application of a transient 3D parallel code PDAC, able to simulate the dynamics of the collapse of the volcanic column and the propagation of the associated PDCs (Esposti Ongaro et al., 2007). The model solves the fundamental multiphase flow transport equation of Neri et al.

(2003) and provides a means to describe the complexities of the column collapse and the temporal and spatial evolution of flows over the whole 3D topography of the volcano. However, the full ranges of plausible volcanic and other physical input parameter variations are not amenable to comprehensive exploration in a restricted number of scenario runs, which are limited by computing power and cost. Under these circumstances, the few PDAC runs that were possible were used as indicative reference simulations, with expert elicitations used to derive rational, quantitative statements about the most

appropriate values to use for variables of interest and, more importantly, to give expression to the scientific uncertainty that attaches to the outcomes of such model runs. For instance, distributional expressions for uncertainties on pyroclastic flow run-out distances, peak pressures and temperatures were obtained by elicitation, after detailed consideration of the few simulation model results that were achievable, and of field evidences, old and new.

One significant source of epistemic uncertainty in this context is the role that the actual topography of Somma-Vesuvius will play in the occurrence of a future central eruption from the present Gran Cono of Vesuvius. After analysing several different options, the EXPLORIS group envisaged a subdivision of the Vesuvian Area into two main sectors, Sectors A and B, delimited by the two red lines on Figure 4A (Sector A includes the area "not protected" by Mt. Somma, and Sector B, the area which is "protected" - representing a first-order source of epistemic uncertainty in respect of the extent to which the

presence of the Mt. Somma topography could determine which areas could be invaded by flows, or modify properties of the flows that might affect the two sectors. More detailed analysis of modelled effects within Sector A suggested a sub-division into Sectors A1, A2, A3 and A4, as delineated by the yellow lines on Figure 4A, with the aim of reducing overall epistemic uncertainty in relation to directional influences on PDC propagation, by allowing more precise analysis of the spatial hazard in the region not protected by Mt. Somma. The bracketed values in each sector show elicited modal probabilities that a PDC





will affect that sector, given a Sub-Plinian I scale eruption occurs (probabilities expressed in percentage terms) together with the corresponding credible intervals, in quantile form [5th, 50th, 95th percentiles]. From the elicitation outcomes it is evident that the presence of Mt. Somma is expected to reduce, by a factor of about two, the probability of flow invasion into the northern sector. Nevertheless, the associated credible interval results are quite large, reflecting the significant uncertainty
associated with judgments about column collapse and PDC phenomena. That said, the elicited probabilities of invasion of the different sub-sectors of Area A are each very similar, and apparently only weakly affected by the preferential propagation directions shown by some of the 3D simulations (Esposti Ongaro et al., 2008) or by reconstructions of past Sub-Plinian events (Rosi et al., 1993; Cioni et al., 2008).

Even more critical information, relating to PDC hazards and hence risk mitigation, is represented in the assessed run-outs of PDC, which directly determine the extent of the Emergency Plan Red Zone, i.e. the size of the region that should be evacuated in advance of an eruption, and where about 500,000 people currently live. Figure 4B shows the elicited judgements of maximum run-out distances (in km) for PDCs occurring during a Sub-Plinian I eruption, by sector. Inner arcs (blue) are 95% confidence levels for a run-out exceeding the distance shown (e.g. 2.5 km for Sector A1), central arcs (green)
are the modal (50th percentile) values, and outer arcs (orange) are the run-out distances assessed has having only a 5% chance of being exceeded (e.g. 13.3 km for A1). A significant difference in anticipated run-outs is again shown for Sectors A and B (a gap of about 2 km between the 50th percentiles), but perhaps the most striking – and important - feature of the results is the large credible intervals associated with these run-out estimates. This outcome was actually expected, given the complexity of the phenomenon being investigated and recognition of the technical limitations of the approaches adopted. In
fact, the mechanism and degree of column collapse, i.e. the percentage of mass collapsing back to the ground, can significantly affect the mobility and dispersal of PDCs. On the other hand, reconstructions of the maximum extent of PDCs that occurred during past events are limited by the incomplete preservation of the products, as well as by partial access to the deposits (Cioni et al., 2008). Similarly, the adopted PDAC 3D code is limited by the vertical resolution of the computational grid, which does not allow accurate modelling of the lower denser portion of the flow (Esposti Ongaro et al., 2007, 2008).

The large epistemic uncertainties regarding the directional controls on PDC probabilities and likely run-outs also influence the expected values of the main physical variables that can be associated with a PDC scenario: e.g. peak dynamic pressure and peak flow temperature. The fact that the EXPLORIS exercise also resulted in large credible intervals associated with these parameter estimates, as well as with the PDC run-outs, clearly reflects expert perceptions of the significant degree to
which epistemic uncertainties must affect current attempts to forecast the complex hazard processes being considered. One conclusion is that more field and more numerical work is needed in order to further constrain the areas likely to be affected by future PDCs at Vesuvius.

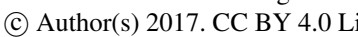



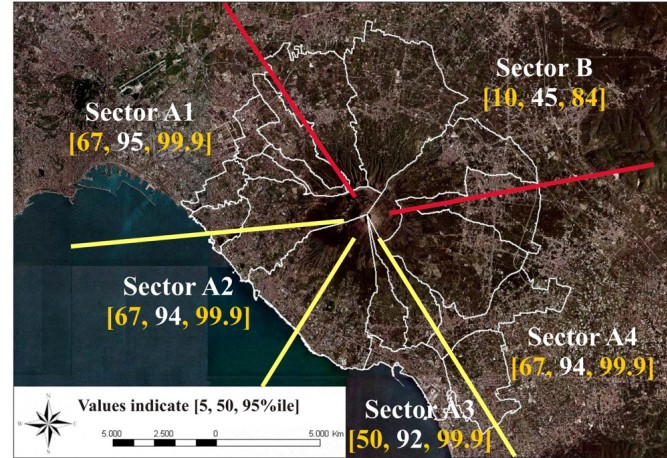

A

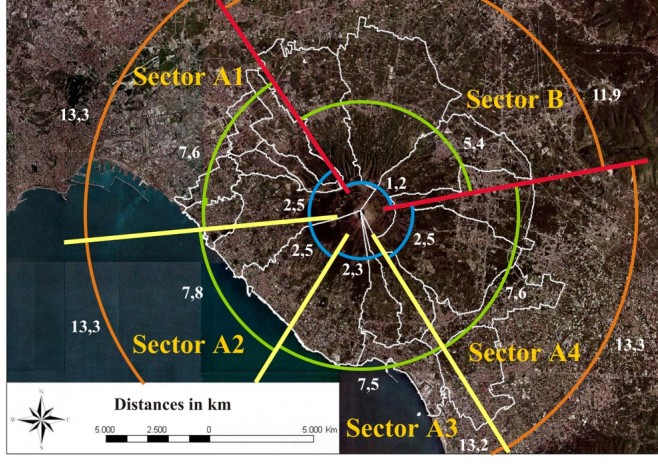

B

**Figure 4. A. Broad segmentation of area around Vesuvius recognising the first-order effect of Mt. Somma topography in determining areas that might be invaded by pyroclastic density current flows (PDC) as the result of a Sub-Plinian I eruption. The bracketed values in each sector show elicited modal probabilities that a PDC will affect that sector (expressed in percentage terms) together with the corresponding credible intervals, in quantile form [5th, 50th, 95th percentiles]. B. Elicited estimates of maximum run-out distances (in km) for PDCs occurring during a Sub-Plinian I eruption, by sector. Inner arcs (blue) are 95% confidence levels for exceeding distance shown (e.g. 2.5 km for Sector A1), central arcs (green) are expected (50th percentile) values, and outer arcs (orange) are the run-out distances assessed has having only a 5% chance of being exceeded. (from Neri et al., 2008, with permission).**





### 9. Windstorms

#### 9.1 Windstorms and key epistemic uncertainties

Weather hazards are a major source of societal risk causing death, destruction to infrastructure and disruption to transport

and business. Global insured losses due to windstorms, currently estimated to cost $2.7 billion annually (Podlaha et al., 2017), are expected to rise dramatically due to climate-change related trends in weather extremes, increasing exposure in developing countries, and increasing world population. Extra-tropical cyclones (also known as *windstorms*) are major contributors to this impact e.g. insured losses in Europe of $9 billion for windstorm Daria (25/1/1990). Furthermore, windstorms often arrive in close succession, which enhances the risk of large aggregate losses e.g. the winter 2013/14 cluster

of European windstorms Christian, Xavier, Dirk and Tini caused insured losses of $1.38, 0.96, 0.47 and 0.36 billion totalling $3.3 billion (source: www.perils.org).  Epistemic uncertainties in the estimation of windstorm risk stem largely from the (poorly supported) choices that have to be made during hazard and impact estimation.

#### 9.2 Uncertainty quantification in windstorm hazard estimation

Windstorm loss distributions are inferred from historical weather measurement data (mainly available since 1950) and also increasingly from storm data simulated *ab initio* from numerical weather and climate prediction models (Schwierz et al. 2010; Pinto et al. 2010; Della-Marta et al. 2010; Renggli et al. 2011; Karremann et al. 2014). The loss distributions are estimated by Monte Carlo simulation using ad hoc combinations of various statistical, dynamical and engineering type models: statistical models for estimating trends and correcting inhomogeneities in the historical data (Barredo 2010), either

low-order parametric stochastic models (the traditional basis of many catastrophe models), or more recently, numerical weather and climate models for simulating large sets of artificial hazard events, statistical models for adjusting biases in numerical model output, and stochastic models for simulating losses from the artificial windstorm events (e.g. compound-Poisson event-loss table models).

Since many choices are required to develop these models, there are many sources of epistemic uncertainty. To list just a few

of the major uncertainties in each type of model:

- Stochastic hazard and loss models often use highly idealised non-physical description of complex storm processes (e.g. polynomial representation of storm tracks). There is the possibility of over-fitting to the data available from relatively short historical periods.  There are often overly restrictive assumptions in simulating losses e.g. homogeneity in time, independence of events, independence of frequency and severity;

- Statistical models require distributional assumptions e.g. extreme value models (Brodin and Rootzén 2009; Della-Marta et al. 2009), assumptions about model-dependence of simulated storms (Sansom et al. 2013), and assumptions about dependency in space-time and between events (Bonazzi et al. 2012; Economou et al. 2014);



- • Numerical weather and climate models show biases in storm properties that have resisted model improvements over the past 40 years e.g. too zonal storm tracks over W. Europe (Zappa et al. 2013), poor representation of small horizontal scale processes even at very high resolution e.g. wind gusts (Ólafsson and Ágústsson 2007), missing processes e.g. sting jets caused by mesoscale features such as stratospheric intrusions (Catto et al. 2010) and non-
adiabatic forcing of storms by anomalous oceanic conditions (Ludwig et al. 2014).

Finally, there is also a major overarching source of epistemic uncertainty in how these different model components should be coupled together. At present there is no accepted theory for how one should and should not do this.

Clustering of windstorms provides a good example of an epistemic uncertainty that has recently received much attention and thereby led to model developments. Analysis of historical reanalysis data revealed that windstorm modulation by large-scale
climate modes leads to more clustering over Europe than one can expect by chance i.e. from an homogeneous Poisson process (Mailier et al. 2006). Furthermore, clustering was also found to increase for more extreme wind speeds (Vitolo et al. 2009), in contradiction to the assumption often made by actuaries suitable for identically distributed variables. This research raised much awareness about clustering in the natural catastrophe insurance industry that has led to major developments in windstorm catastrophe models (Khare et al. 2014). The findings are also stimulating new research into mechanisms for
clustering of extreme storms (e.g., Rossby wave breaking; Pinto et al., 2014).

## 10. Co-emergent and Cascading Hazards

The earlier discussion has mostly been concerned with the characteristics of individual hazards but it is clear that an assessment of risk often needs to allow for the joint occurrences of cascading multiple hazards, either for hazards of different
types affecting a single location, or the joint occurrence of a hazard at multiple locations simultaneously (e.g. Lamb et al., 2010; Gill and Malamud, 2014; Keef et al., 2013; De Risi and Goda, 2016; Goda et al., 2016). Both will affect the assessment of the joint risk. In some cases the joint risk may be causative, including the dependence of numerous aftershocks triggered by a main shock (Yeo and Cornell, 2009); tsunamis initiated by ocean floor earthquakes and landslides (Tappin et al., 2014; Goda et al., 2016); the landslide and avalanches that result directly from earthquakes; and the potential
for landslide as well as flood impacts on dam safety (an epistemic uncertainty that is usually neglected but which has caused past dam overtopping). In other cases independent occurrences might contribute to an increased risk, such as the joint occurrences of fluvial floods, high tides and atmospheric surge on the risk of estuarine and coastal flooding. Assessing the joint frequency of such events has been receiving increasing attention (e.g. Svensson and Jones 2004). In particular, the covariation of different causes of the hazard, and joint occurrences across multiple locations has been investigated using
flexible functional relationships based on overlap likelihood relationships (Gill and Malamud, 2014) and copulas (e.g. Keef et al., 2013). An interesting application of the latter was used to produce the probabilistic flood map of Figure 2 in Section 2.2, which is affected by the joint occurrences of high flows both in the mainstream river and two major tributaries entering from the south (Neal et al., 2013).



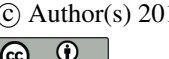

### 11. Uncertainty quantification related to the consequences of natural hazards

Alongside uncertainties related to the characterization and propagation of the hazard itself (e.g. the footprint and magnitude of an earthquake), risk assessments also entail epistemic uncertainties arising from the uncertain consequences and damages of the hazard (i.e., the loss part of the risk assessment). Key components of such assessments are (Tesfamariam and Goda,
2013): exposure (e.g. spatial locations of populations and assets), vulnerability (e.g. characteristics of buildings and infrastructure), and loss (e.g. characteristics of assets and loss generation mechanisms). All these involve significant uncertainties. In the risk equation (i.e. convolution of hazard, exposure, vulnerability, and loss), these uncertainties are propagated and integrated. Uncertainties in exposure and loss are attributed to a lack of information, incomplete knowledge,
as well as simplification adopted in the models, and thus are largely epistemic.

The consequences of hazards are often difficult to quantify and there is still little research available linking the characteristics of the hazard (e.g., e.g. drought duration and severity) to the related consequences (e.g. Jenkins 2013). For past events there might be some epistemic uncertainty about the damages associated with the event, but there is often
considerable uncertainty about what is actually at risk, i.e. the exposure (e.g. Chatterton et al., 2014). Damages that are covered by insurance are generally well known (but subject to commercial confidentiality restrictions and not readily available), but, not all damages are insured and not all are easily expressed in monetary terms (such as damage to habitats, cultural heritage, and loss of life). Indirect damages to businesses and individuals (e.g. as a result of infrastructure failures, health and psychological impacts) can also be difficult to assess. More geographically explicit damage relationships are
needed for hazards such as droughts (Bachmair et al., 2015; Blauhut et al., 2015) or tsunamis (Goda and Song, 2016), which cover potentially large and heterogeneous areas. Potential sources of damage are even more difficult to estimate for future events, as a result of epistemic uncertainties (e.g. about policy changes in flood risk management, planning decisions for flood plain developments, changes in availability of insurance cover, etc). Different chosen loss models might result in quite different estimates of the consequences of an event (e.g. Jongman et al., 2012; Chandler et al., 2014), to the extent that
estimates of risk might generate significant controversy as a result of the epistemic uncertainties inherent in the assessment processes (e.g. Penning-Rowsell, 2015).

Epistemic uncertainties in the risk assessment of natural hazard consequences also arise from the interactions of the hazard with human actions. Consider the example of droughts. Because of their temporal and spatial extent, droughts are more
prone to mitigation or exacerbation by socio-economic drivers than some other natural hazards. Those responding to or managing water resources during drought will make use of nearby water resources or stored water, thus actively intervening to influence the development and consequences of the event (Van Loon et al., 2016a). For instance, epistemic uncertainties arise from incomplete knowledge of how demand responds during times of drought to both environmental conditions (weather) and management actions (i.e., water use restrictions, price increases) (Kenney et al., 2008). Although hot/dry



weather may increase demand in the short-term, it is not clear which climatic variables are best suited to explain water consumption patterns (Kenney et al., 2008). Over larger spatial and temporal scales, changes in water demand are difficult to project and add a level of epistemic uncertainty to any water resources planning decision. Water managers often rely on extrapolation processes (Jorgensen et al., 2009; House-Peters and Chang, 2011), yet this process has not been entirely

successful, with the UK's largest reservoir at Kielder built to meet projections which did not foresee the decline in heavy industry in the North of England (Walker, 2012), a clear case of the impact of epistemic uncertainty about future boundary conditions but which, opportunely, has served to mitigate the effects of drought in the area. This type of uncertainty, combined with data gaps, makes the modelling tools available largely inadequate to predict drought impacts. Severe limitations exist in predicting the impacts of feedbacks and modifications to drought events due to human actions, calling for

a new framework for drought risk assessment that includes the human role in mitigating (or enhancing) the consequences of drought (Van Loon et al., 2016b).

## 12. Generalisations across hazard areas

In reviewing the way in which epistemic uncertainties are handled in each of these natural hazard areas, certain

commonalities are apparent.  Most notable is the tendency for treating all sources of uncertainty as aleatory variables, for both the hazard and the consequences or impacts that make up the risk equation.  In most hazard areas, probabilistic methods are replacing older deterministic probable maximum event methods.  The probabilistic approach is attractive in that the power of statistical theory, including the use of judgement-based probabilities in a Bayesian framework, can be utilized. However, when used to represent epistemic uncertainties such an approach will be subject to limitations that include:

20       • not allowing for the incompleteness of probability assessments (including the probabilities associated with the branches of logic trees)
         • the potential of over-fitting to limited historical records in estimating the frequencies of extreme events of unknown (and potentially non-stationary) distributional form, and
         • the limitations of expert elicitation of prior probability and scenario information.

This suggests that an extension to a more explicit recognition of epistemic uncertainties might be necessary in future. Opinions on how to do so are discussed in Part 2 of this paper.

## 13. Conclusions

This paper has reviewed examples of how uncertainties in general, and epistemic uncertainties in particular have been

handled in assessments of risk associated with different natural hazards. In most cases, epistemic uncertainties are not considered explicitly, but are still treated as if they can be considered as aleatory variables of specified distributional form. This almost certainly leads to an underestimation of the uncertainty in the risk assessment, and might lead to a lack of robustness of decision and to future surprise.  It is therefore both possible and desirable to extend the analysis to explicitly include different scenarios of epistemic uncertainty. The analysis of different natural hazard areas presented above makes it

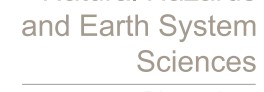



clear that there are different degrees of appreciation for and approaches to dealing with epistemic uncertainty. We hope that making this comparison will enable researchers in different areas to learn about structured approaches that are being used elsewhere, particularly in dealing with uncertainties that are less amenable to being treated probabilistically.

Where observational data are available that can be used to constrain the prediction uncertainties in an application, then care should be taken in the form of model evaluation.   Treating a residual series as a simple aleatory variable can be used to define a formal statistical likelihood function, but if the uncertainties are dominated by epistemic sources the result may be overconfidence in model selection and over-constraint of the predictive uncertainty.  In that case of real-time forecasting, data assimilation can be used to adaptively compensate for unknown uncertainties in improving forecasts and constraining

forecast uncertainties over the lead times of interest, at least where the data can be processed within the time scale of the system response or at a temporal resolution useful to decision makers.

The variety of assumptions and approaches being used in different hazard application areas reinforce the discussion that follows in Paper 2 about the importance of a framework for structured analysis and communication of the assumptions and

the meaning of an uncertainty analysis, particular to the decision makers and other users.  There is no single way of assessing the impacts of epistemic uncertainties on risk (for good epistemic reasons), but in encouraging good practice we can at least demand clarity in the assumptions that are made, with the possibility that this then might lead to some consideration of alternative assumptions and a consequent reduction in the potential for future surprise.

**Acknowledgements**

This work is a contribution to the CREDIBLE consortium funded by the UK Natural Environment Research Council (Grant NE/J017299/1).  Jeff Neal and Dave Leedal are thanked for their work in producing Figure 2.

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
