# Peer review of "Epistemic uncertainties and natural hazard risk assessment. A review of different natural hazard areas"

_Natural Hazards and Earth System Sciences, 2017_

## Referee Comment (RC1) · Anonymous Referee #1 · 6 Sep 2017

Epistemic uncertainties and natural hazard risk assessment 1. A review of different natural hazard areas
Beven et al.

Given my specific area of expertise I have focused my review on the abstract, introduction, section 7 (volcanic eruptions and ash clouds) and the conclusion. I'm afraid that I could not tally the authors responses to my previous comments (linked from the initial submission) with the text in this new submission so have treated it as a new paper.

This paper provides an overview of the major sources of epistemic uncertainty in several different natural hazard areas. The review is broken down into hazard areas and a very short section is included towards the end identifying common approaches. Whilst the paper includes a fairly good review of the recent literature, overall, I found it hard to determine if the paper has achieved its main aim as I was not sure what the main aim of the paper was (see comments below). Therefore, I recommend major corrections.

Comments
1. The abstract is rather short and missing some key information. For example, what is the motivation for the work and the main aim. Also, what methods have been employed to achieve the aim? I appreciate from the title that it is a review paper focussing on epistemic uncertainties but it was not obvious from the abstract why this review was undertaken and what the authors aim to achieve by the review.
2. From the introduction, it is not clear what the aim of the review paper is. If it is to identify common approaches between the hazards then the generalisation section (section 12) is far too brief. If it is to explain the difference between the treatment of aleatory and epistemic uncertainty in the different hazard areas then it needs to be more explicit. Are the authors suggesting that all of the epistemic uncertainties they list in section 7 are currently treated incorrectly in the literature (as aleatory uncertainties) and that this could have large consequences? This is implied in the conclusion at least.
3. In the introduction, the authors make the point that uncertainty estimation in some hazards (including volcanic ash?) 'returns zero results', yet in section 7 they refer to many papers that attempt to quantify or to reduce this uncertainty so I was confused by this apparent contradiction.
4. I found the lack of structure in section 7 slightly confusing. As far as I can tell this section includes a review of:
   a. Uncertainties in satellite retrievals (p23, l14-34 and p25, l1-8).
   b. Uncertainties in numerical modelling (p25, l10-34 and p26, l1-4).
   c. Using satellite retrievals to constrain uncertainty (p26, l6-31).
   d. Using ensembles to quantify uncertainty (p26, l32-335 and p27, l1-3)
   e. Operational challenges? (p26, l5-11).
   Inclusion of these subsections (or something similar) would help to guide the reader through the section.
   4.1 For (a) why only focus on satellite retrievals and ignore a discussion of retrievals from ground-based and airborne in-situ measurements of volcanic ash? If this is in the interests of space then this omission should be justified in some way.
   4.2 For (b) uncertainties associated with the eruption source parameters and missing processes are covered in some detail but the discussion of uncertainties associated with the representation of physical processes within numerical models (e.g. advection, dispersion, removal) is described in a single sentence (p26, l1-2)

which seems a bit un-balanced given the large sensitivity of volcanic ash concentrations to the magnitude of turbulence and wet deposition used in models.

4.3 For (c) and (d) these sections seem to offer the best insight into how epistemic uncertainties can be quantified and potentially reduced for ESPs and advection at least. Perhaps you could also refer to Harvey et al. (2016) for an example of the treatment of dispersion and removal processes also?

4.4 For (e) is the purpose of this section to demonstrate that currently none of these uncertainties are accounted for in the operational setup? This was not clear to me.

Harvey, N., Huntley, N., Dacre, H., Goldstein, M., Thomson, D. and Webster, H., 2016. Multi-level emulation of a volcanic ash transport and dispersion model to quantify sensitivity to uncertain parameters. *Nat. Hazards Earth Syst. Sci. Discuss.*

---

## Referee Comment (RC2) · Anonymous Referee #2 · 12 Jan 2018

The manuscript gives an overview of the type of epistemic uncertainties encountered in various scientific disciplines related to natural hazards. I understand that most of the analysis and discussion takes place in the companion paper (part 2), which necessarily makes this paper more descriptive in nature. Nevertheless, the current version reads very much as an exhaustive, but also very exhausting, laundry list of all the "unknowns" that the authors have collectively been able to identify.

In itself, the paper is well written (indeed I could not find anything related to language or spelling to comment on) but nevertheless I found it very tedious to read. This is not only because of the length (at some point it almost felt like an endless lament of things

that we don't know yet), but also because no reader can be expected to be an expert in all the processes and methods that are mentioned and discussed, and thus easily gets lost (or loses interest). I suggest two points of action to remediate this:

(1) provide a clear framework for analysis. This would greatly improve the structure and systematic nature of the review.

(2) reduce the length dramatically (by perhaps a third or even a half).

With regard to (1), I think that a much more elaborate theoretical framework of thinking about uncertainties would be very helpful. The distinction between epistemic and aleatory uncertainties is useful, but each category is still a very broad umbrella for uncertainties of a very different nature. Surely much more fine-grained classifications and distinctions exist and can be used to structure the review? I do not consider myself an expert on the theory of uncertainty (instead more of a practitioner in one of the covered fields, but therefore probably representative for much of the journal's readers). But while reading the long list of epistemic uncertainties, I could not help but feel that these are still very diverse in nature, with very different issues and bottlenecks. For instance, sometimes it seems to be a well understood process that simply suffers from a severe lack of data (which is clearly endemic in all covered areas). Perhaps sometimes a probabilistic model may exist in theory, but no analytical representation to conceptualize it. Sometimes it may simply be the modeller being sloppy (or insufficiently conservative) on the implications of certain assumptions behind a probability model. Sometimes models are used for mathematical convenience or computational necessity (e.g., the Gaussian model) rather than a true belief that it fits perfectly the nature of the phenomenon...

This is not a suggestion for classification (again I am not an expert) but simply some examples to highlight that lumping error sources under just two large umbrellas may leave the reader unguided (and unsatisfied). It also may not do justice to an undoubtedly large body of literature on the nature of uncertainty. For instance, surely within

the statistical community there must be a lot of thinking about the consequences of incorrect models? Again lumping everything in "epistemic" vs "aleatory" (which reads very much as "non-statistical" vs. "statistical") would not seem to do this justice.

Again, I suppose that much of the second paper is dedicated to this type of discussion. But now it feels that simply all the raw review material is presented in this paper, before the second paper tries to make sense of it. I don't think that this is ideal, and I do think that a lot can be done to alleviate it by taking a more systematic and structured approach from the start.

Much of this would also solve issue (2), in addition to some more rigorous editing (by the authors, not the editor, that is). For instance, quite some space is dedicated to arguing the societal impact and relevance of the hazards (e.g., P9/L9-19). As interesting as this is in its own right, it is probably not truly relevant for the argument of this paper.

Specific comments:

P1/L22: "It is suggested...": I wonder whether this sentence is refers to this paper or perhaps to the companion paper? The manuscript indeed mentions the use of scenario analysis in various disciplines, but there is no explicit discussion or argumentation of this, except for a single sentence in the conclusions, which mentions that it is "possible and desirable to extend the analysis to explicitly include different scenarios of epistemic uncertainty", but this comes out of the blue.

P8/L20: "base [...] based": revise sentence construction?

---

## Referee Comment (RC3) · R E Chandler (Referee) · 29 Jan 2018

I have today received the final report from the invited reviewers. It was sent to me by email because there seemed to be a glitch with the Copernicus system. The reviewer comments are pasted below, therefore.

Richard

«« STARTS This paper covers a wide range of different hazards, but in a series of sections that bear little relation to each other, do not make good use of the framework provided by the classification or hierarchy of uncertainties in the introductory section;

[Figure]

and furthermore are not brought together at the end in a comparative analysis of the problems of different sub-disciplines of geohazards, or in a discussion of how methods from one area could be applied to others. Since the authors were part of a large research programme, one of whose aims was to do these things, this failure is doubly disappointing.

The classification of uncertainties in the introductory section into aleatory, epistemic and ontological is potentially very useful in my view, but the authors then don't follow it through as whilst epistemic uncertainties in each area are admitted in the respective sections (for example, in the determination of regional maximum earthquake magnitudes in the seismological hazard section, and the question of Poissonian vs. non-Poissonian earthquake recurrence), no-one seems willing to acknowledge ongoing ontological uncertainties (aka ongoing scientific revolutions in the Kuhnian sense?) in their particular sections. Again, an example from the seismological field: there is no mention at all in this manuscript of the recent work of Stein, Geller, Mulargia, Stark and colleagues that questions the whole foundations of PSHA. See for example:

Mulargia, F., Stark, P. B., & Geller, R. J. (2017). Why is probabilistic seismic hazard analysis (PSHA) still used?. Physics of the Earth and Planetary Interiors, 264, 63-75.

Liu, M., & Stein, S. (2016). Mid-continental earthquakes: Spatiotemporal occurrences, causes, and hazards. Earth-Science Reviews, 162, 364-386.

Stein, S., Geller, R. J., & Liu, M. (2012). Why earthquake hazard maps often fail and what to do about it. Tectonophysics, 562, 1-25.

Stark, PB (2016) Pay no attention to the model behind the curtain (online at https://pdfs.semanticscholar.org/7fda/9700ceb2e34c7d0a8720a17a099d5e273111.pdf)

Whatever one thinks of the balance of the argument between the advocates of classical PSHA and these iconoclasts, it seems to me that the debate between these two groups provides as fine an example of ontological uncertainty in hazard assessment as exists

today, so it is deeply disappointing that the seismic hazard section does not make use of this controversy, if only as an example of ontological uncertainty.

Another problem with the lack of connection between sections of the paper is that it does not therefore provide a good analysis of how uncertainties can propagate between analysis of related hazards: thus, for example, probabilistic tsunami hazard analysis contains very large uncertainties due to the uncertainties in the understanding of the occurrence distributions of the causative events (mainly earthquakes, so if PSHA is fatally flawed as Stein et al argue, then so is PTHA). Similarly, connections and feedbacks exist between flood hazards and (rainfall-triggered) landslides and also extreme winds associated with intense rainfall since debris from landslides and wind-toppled trees entering rivers during extreme flood events, can exacerbate the flooding by blocking flow under bridges and through narrow channels, causing overbank flooding (see the recent examples from Dominica during Hurricane Maria in September 2017). Such feedbacks again require uncertainties to be propagated from one set of models into others, and this is another challenging area with large epistemic and ontological uncertainties that are not adequately covered in this review.

A final topic that needs more consideration is that of how the importance of different types of uncertainty varies according to the practical use to which the modelling approach concerned is put. Although the authors of some sections of the paper do make the distinction between probabilistic and real-time warning approaches to hazard mitigation in their respective sections, the implications of this need to be explored further in terms of how tolerant different mitigation strategies are of different types of uncertainty in hazard estimates. Thus, permanent mitigation strategies (in the sense defined by Day & Fearnley, 2015) are extremely sensitive to uncertainties in probabilistic hazard analyses especially at the high-intensity range where the effectiveness of strategies such as building construction codes are liable to break down; whereas responsive and anticipatory mitigation strategies (e.g. tsunami evacuations and volcanic eruption warnings, respectively) are less sensitive to probabilistic uncertainties but are

highly dependent on accurate and timely detection and quantification of specific hazard events.

[reference: Day, S.J., & Fearnley, C. (2015). A classification of mitigation strategies for natural hazards: implications for the understanding of interactions between mitigation strategies. Natural Hazards, 79(2), 1219-1238.]

Overall, the paper contains some useful material but it is not properly examined or brought together (which is the whole point of "review" or "personal perspective" papers) so I do not think that it is suitable for publication in its present form and needs further substantial revision and re-review.

»» ENDS

---

## Author Comment (AC1) · 16 Apr 2018

Response to Reviewers of "**Epistemic uncertainties and natural hazard risk assessment.  1. A review of different natural hazard areas**"

**Reviewer 1 – FOCUS ON VOLCANIC HAZARDS AND ASH CLOUDS**

Given my specific area of expertise I have focused my review on the abstract, introduction, section 7 (volcanic eruptions and ash clouds) and the conclusion. I'm afraid that I could not tally the authors responses to my previous comments (linked from the initial submission) with the text in this new submission so have treated it as a new paper. This paper provides an overview of the major sources of epistemic uncertainty in several different natural hazard areas. The review is broken down into hazard areas and a very short section is included towards the end identifying common approaches. Whilst the paper includes a fairly good review of the recent literature, overall, I found it hard to determine if the paper has achieved its main aim as I was not sure what the main aim of the paper was (see comments below). Therefore, I recommend major corrections.

*Comments*

1. The abstract is rather short and missing some key information. For example, what is the motivation for the work and the main aim. Also, what methods have been employed to achieve the aim? I appreciate from the title that it is a review paper focussing on epistemic uncertainties but it was not obvious from the abstract why this review was undertaken and what the authors aim to achieve by the review.

Response: We will extend the abstract to better discuss this manuscript, its content and aims, as well as its connection to the companion paper. In principle, this paper is the review part for the second paper in which all of the similarities and dissimilarities between hazard areas are discussed and a common framework is approached. Paper 1 is not meant to stand on its own but is to be read in connection with paper 2. Paper 1 is a literature review and hence there is no formal method apart from reviewing the literature.

2. From the introduction, it is not clear what the aim of the review paper is. If it is to identify common approaches between the hazards then the generalisation section (section 12) is far too brief. If it is to explain the difference between the treatment of aleatory and epistemic uncertainty in the different hazard areas then it needs to be more explicit. Are the authors suggesting that all of the epistemic uncertainties they list in section 7 are currently treated incorrectly in the literature (as aleatory uncertainties) and that this could have large consequences? This is implied in the conclusion at least.

Response: It is important to view this paper as part 1 of a two-part paper and not as a complete paper by itself. This paper reviews the literature across hazards to identify common approaches as well as common omissions in the treatment of epistemic uncertainty. The 'generalization section' if you like is the companion paper in which an umbrella is created bring the diversity (or lack thereof) together in a common discussion. We went for a two-paper structure because it would have been infeasible to do both in a single paper. We will make clearer at the end of paper 1, how the generalization is achieved in paper 2.

3. In the introduction, the authors make the point that uncertainty estimation in some hazards (including volcanic ash?) 'returns zero results', yet in section 7 they refer to

many papers that attempt to quantify or to reduce this uncertainty so I was confused by this apparent contradiction.

Response: We will delete the figure and take out this brief reference to what has been published here. The issue is one of finding the correct keywords to find all relevant literature via Web of Science. The issue will be dealt with in the individual hazard sections.

4. I found the lack of structure in section 7 slightly confusing. As far as I can tell this section includes a review of:
a. Uncertainties in satellite retrievals (p23, l14-34 and p25, l1-8).
b. Uncertainties in numerical modelling (p25, l10-34 and p26, l1-4).
c. Using satellite retrievals to constrain uncertainty (p26, l6-31).
d. Using ensembles to quantify uncertainty (p26, l32-335 and p27, l1-3)
e. Operational challenges? (p26, l5-11).
Inclusion of these subsections (or something similar) would help to guide the reader through the section.

Response: Yes, the section is slightly long without sub-sections. The issue is that we decided to have similar structures for each of the sections, rather than different structures for each of the hazards addressed. We proposal to include sub-headings for clarity without adding more sub-heading numbers. Hence the structure at the higher level remains while information to guide the reader is available. Thank you for the suggested sub-headings, which we will build on in the revised version of the paper.

4.1 For (a) why only focus on satellite retrievals and ignore a discussion of retrievals from ground-based and airborne in-situ measurements of volcanic ash? If this is in the interests of space then this omission should be justified in some way.

Response: We agree and will balance this out by adding some additional text while reducing the one that is currently there.

4.2 For (b) uncertainties associated with the eruption source parameters and missing processes are covered in some detail but the discussion of uncertainties associated with the representation of physical processes within numerical models (e.g. advection, dispersion, removal) is described in a single sentence (p26, l1-2) which seems a bit un-balanced given the large sensitivity of volcanic ash concentrations to the magnitude of turbulence and wet deposition used in models.

Response: We agree and will balance this out by adding some additional text while reducing the one that is currently there.

4.3 For (c) and (d) these sections seem to offer the best insight into how epistemic uncertainties can be quantified and potentially reduced for ESPs and advection at least. Perhaps you could also refer to Harvey et al. (2016) for an example of the treatment of dispersion and removal processes also?

Response: Yes, that is a good suggestion and we will include Harvey et al. in the suggested manner.

We will include the published version of this paper: *Harvey, N., Huntley, N., Dacre, H., Goldstein, M., Thomson, D. and Webster, H., 2018. Multi-level emulation of a volcanic ash transport and dispersion model to quantify sensitivity to uncertain parameters. Nat. Hazards Earth Syst. Sci., 18, 41-63.*

4.4 For (e) is the purpose of this section to demonstrate that currently none of these uncertainties are accounted for in the operational setup? This was not clear to me.

Response: The idea of this section was to discuss the difference between implementations for research purposes and practical implementations in operational settings.

**Reviewer 2**

The manuscript gives an overview of the type of epistemic uncertainties encountered in various scientific disciplines related to natural hazards. I understand that most of the analysis and discussion takes place in the companion paper (part 2), which necessarily makes this paper more descriptive in nature. Nevertheless, the current version reads very much as an exhaustive, but also very exhausting, laundry list of all the "unknowns" that the authors have collectively been able to identify.

Response: We fully agree that this is a long paper with much detail. It is difficult to balance the need for depth and detail, with that of brevity. We will revise the content of paper 1 and look for opportunities to shorten it now that paper 2 has been accepted.

In itself, the paper is well written (indeed I could not find anything related to language or spelling to comment on) but nevertheless I found it very tedious to read. This is not only because of the length (at some point it almost felt like an endless lament of things that we don't know yet), but also because no reader can be expected to be an expert in all the processes and methods that are mentioned and discussed, and thus easily gets lost (or loses interest). I suggest two points of action to remediate this:
  (1) provide a clear framework for analysis. This would greatly improve the structure and systematic nature of the review.
  (2) reduce the length dramatically (by perhaps a third or even a half).

Response: We fully agree that the paper is too long – even though reducing it by half would be excessive. We will try to shorten it without losing the relevant discussions of each hazard area.

The other point the reviewers makes refers to the purpose of this paper and how we assume readers will use it. In contrast to paper 2, we do not think that this paper is meant to be read from front to back in one go. We also certainly do not assume that readers are experts in all of these areas – who could be? Rather, we review a wide range of hazard areas that are generally not placed next to each other but treated in separate papers or even in separate journals. We assume that readers will in general be experts in one of the areas discussed, which might be their logical starting point. They can then use this paper interactively with paper 2 to see how their own area relates to other hazard areas of interest, e.g. with the aim of building connected hazard models. Hence, we do not think that the exhaustive aspect of the paper is a large problem (it is of cause a problem for a reviewer who is asked to comment on all of the paper).

With regard to (1), I think that a much more elaborate theoretical framework of thinking about uncertainties would be very helpful. The distinction between epistemic and aleatory uncertainties is useful, but each category is still a very broad umbrella for uncertainties of a very different nature. Surely much more fine-grained classifications and distinctions exist and can be used to structure the review? I do not consider myself an expert on the theory of uncertainty (instead more of a practitioner in one of the covered fields, but therefore probably representative for much of the journal's readers). But while reading the long list of epistemic uncertainties, I could not help but feel

that these are still very diverse in nature, with very different issues and bottlenecks. For instance, sometimes it seems to be a well understood process that simply suffers from a severe lack of data (which is clearly endemic in all covered areas). Perhaps sometimes a probabilistic model may exist in theory, but no analytical representation to conceptualize it. Sometimes it may simply be the modeller being sloppy (or insufficiently conservative) on the implications of certain assumptions behind a probability model. Sometimes models are used for mathematical convenience or computational necessity (e.g., the Gaussian model) rather than a true belief that it fits perfectly the nature of the phenomenon...

Response: There are several things the reviewer refers to here. [1] The reviewer is correct with his examples of the nature of epistemic uncertainty, e.g. lack of data (which is the common problem). We therefore "For the practical purposes of this review, we will define epistemic uncertainty as those uncertainties that are not well determined by historical observations". [2] We initially placed the companion (discussion) paper first, which was then followed by this detailed review. In a sense introducing the theoretical framework that the reviewer is asking for here. However, we (and the reviewers during our first submission a couple of years ago) did not find this a helpful structure and we reversed the order. Here, we therefore give a relatively simple definition and the proceed to look through the different hazard areas for examples of how this lack of determination creates epistemic uncertainties in the different hazard areas. We then derive a theoretical framework (as much as possible) in paper two.

This is not a suggestion for classification (again I am not an expert) but simply some examples to highlight that lumping error sources under just two large umbrellas may leave the reader unguided (and unsatisfied). It also may not do justice to an undoubtedly large body of literature on the nature of uncertainty. For instance, surely within the statistical community there must be a lot of thinking about the consequences of incorrect models? Again lumping everything in "epistemic" vs "aleatory" (which reads very much as "non-statistical" vs. "statistical") would not seem to do this justice.

Response: While we recognise that many classifications of uncertainty are possible, we would argue for simplicity in this respect. Certainly, since we do not know what the "correct" odel should be, model structural uncertainty can be considered and treated as a form of epistemic uncertainty. Regarding the statistical literature, we do not aim to also review the statistical literature, which has also not yet converged to a single approach in this area, but a range of strategies that would make reading our paper even more exhausting. This issue is discussed in some detail in paper 2, whereas here we wanted to focus on what has been implemented in the different hazard areas. We do not think that there is an easy (or ready) classification we could use to structure the area of epistemic uncertainty more without adding more discussion (which would again open for differences in opinion).

Again, I suppose that much of the second paper is dedicated to this type of discussion. But now it feels that simply all the raw review material is presented in this paper, before the second paper tries to make sense of it. I don't think that this is ideal, and I do think that a lot can be done to alleviate it by taking a more systematic and structured approach from the start.

Response: We will try to foresee the second paper in the structure of paper 1 and reduce it somewhat, while anticipating some of the elements better (for better structure). We have gone back and forth many times between the order of the papers and there is no order that is clearly much better (we think). It remains a matter of opinion.

Much of this would also solve issue (2), in addition to some more rigorous editing (by the authors, not the editor, that is). For instance, quite some space is dedicated to arguing the societal impact and relevance of the hazards (e.g., P9/L9-19). As interesting as this is in its own right, it is probably not truly relevant for the argument of this paper.

Response: Yes, there are certainly places where we can trim the paper and we will do so. The suggestions for possible trimming are very helpful and we will use them.

Specific comments:

P1/L22: "It is suggested...": I wonder whether this sentence is refers to this paper or perhaps to the companion paper? The manuscript indeed mentions the use of scenario analysis in various disciplines, but there is no explicit discussion or argumentation of this, except for a single sentence in the conclusions, which mentions that it is "possible and desirable to extend the analysis to explicitly include different scenarios of epistemic uncertainty", but this comes out of the blue

Response: Yes, this formulation is somewhat unfortunate due to the wide usage of the term 'scenario'. We will rephrase this.

P8/L20: "base [...] based": revise sentence construction?

Response: Will be revised.

**Reviewer 3 – FOCUS ON SEISMIC HAZARDS**

This paper covers a wide range of different hazards, but in a series of sections that bear little relation to each other, do not make good use of the framework provided by the classification or hierarchy of uncertainties in the introductory section; and furthermore are not brought together at the end in a comparative analysis of the problems of different sub-disciplines of geohazards, or in a discussion of how methods from one area could be applied to others. Since the authors were part of a large research programme, one of whose aims was to do these things, this failure is doubly disappointing.

Response: We agree that this paper (paper 1) does not achieve both an introduction to the topic, a review across hazard areas, and, ultimately, a bringing together across hazard areas. It did not aim to do so, since Paper 2 sets out to define good practice in the third task (of bringing together).

The classification of uncertainties in the introductory section into aleatory, epistemic and ontological is potentially very useful in my view, but the authors then don't follow it through as whilst epistemic uncertainties in each area are admitted in the respective sections (for example, in the determination of regional maximum earthquake magnitudes in the seismological hazard section, and the question of Poissonian vs. non-Poissonian earthquake recurrence), no-one seems willing to acknowledge ongoing ontological uncertainties (aka ongoing scientific revolutions in the Kuhnian sense?) in their particular sections. Again, an example from the seismological field: there is no mention at all in this manuscript of the recent work of Stein, Geller, Mulargia, Stark and colleagues that questions the whole foundations of PSHA. See for example:

- Mulargia, F., Stark, P. B., & Geller, R. J. (2017). Why is probabilistic seismic hazard analysis (PSHA) still used?. Physics of the Earth and Planetary Interiors, 264, 63-75.
- Liu, M., & Stein, S. (2016). Mid-continental earthquakes: Spatiotemporal occurrences, causes, and hazards. Earth-Science Reviews, 162, 364-386.
- Stein, S., Geller, R. J., & Liu, M. (2012). Why earthquake hazard maps often fail and what to do about it. Tectonophysics, 562, 1-25.
- Stark, PB (2016) Pay no attention to the model behind the curtain (online at https://pdfs.semanticscholar.org/7fda/9700ceb2e34c7d0a8720a17a099d5e273111.pdf)

Response: Before responding to Reviewer 3's comments below, we would like to emphasise three points:
- As stated in Introduction, our aim of this part 1 review paper is: '*to discuss how epistemic uncertainties have been recognised and treated in the different hazard areas*'. The part2 paper attempts to bring together some communalities and differences of epistemic uncertainty modelling across different hazards.
- This paper inevitably cannot cover all important problems related to epistemic uncertainties in seismic hazards, unlike review papers specific to earthquake hazards (e.g. Stein et al., 2012). Moreover, Reviewer 2 strongly suggested to reduce the length of this manuscript. Considering this suggestion, we aim at least not to increase the manuscript length during our revision.
- Although it is interesting to classify the epistemic uncertainty into 'known/modelled' and 'ontological' ones, this is not followed in this manuscript or in the seismic hazard section because we adopted a broad definition of epistemic uncertainties as '*those that are not well determined by historical observations*' in contrast with aleatory uncertainties, which is more applicable to a wide range of natural hazard areas.

We view PSHA as an engineering endeavour, rather than purely scientific one. Prof. C. Allin Cornell was a civil engineer who specialised in structural reliability and safety, and he was the first to propose a particular numerical formulation for PSHA (which reflected computing power constraints in the 1960s). This was a practical framework/tool for determining a set of seismic hazard estimates for aiding the revision and implementation of seismic design in national building codes. Cornell's procedure has evolved only modestly in the intervening decades (as is evident from, for example, McGuire [2001, 2004]). Thus, there exist major challenges related to modern earthquake hazard assessments and still some limitations to the current numerical implementations of PSHA. These are highlighted in our paper at many places: examples of the 2011 Tohoku earthquake and tsunami and the 2010-2011 Christchurch earthquake sequences are mentioned, as was done by, for example, Stein et al. (2012) [note: this paper was originally cited in the manuscript]. In our part 1 review, we do not want to advocate/criticise theories or methods (i.e. not within the scope of our review). Rather we summarise/discuss how epistemic uncertainties in seismic hazard have been treated. For this purpose, PSHA serves as a common starting point for a wider audience.

We are aware that some researchers and practitioners have strong objections to PSHA (e.g. Mulargia et al., 2017). It is not our intention to exclude the view expressed by Mulargia et al. (2017), and we cite and mention this paper in the seismic section of our revised manuscript. Some of the specific examples mentioned by Reviewer 3 (e.g. Poissonian vs. non-Poissonian occurrences) are deep epistemic uncertainties as available data cannot resolve alternative theories/models for some cases, but also due to misapplications of the theories/models and lack of critical evaluations of underlying hypotheses/assumptions. It is not just because PSHA is not perfect; we think that problems are more complex than that. We think it is important to mention 'non-scientific' aspects of seismic hazard assessments, which we tried to highlight through expert elicitation.

Whatever one thinks of the balance of the argument between the advocates of classical PSHA and these iconoclasts, it seems to me that the debate between these two groups provides as fine an example of ontological uncertainty in hazard assessment as exists today, so it is deeply disappointing that the seismic hazard section does not make use of this controversy, if only as an example of ontological uncertainty.

Response: As mentioned above, we include Mulargia et al. (2017) in our revised manuscript, and this paper is mentioned in the seismic section with regard to PSHA. We do not want to frame our discussions as a debate related to whether PSHA is, in some sense, "correct". Rather, using PSHA, we discuss major epistemic uncertainties that should be taken into account when seismic hazards are evaluated. Although Reviewer 3 mainly focuses on epistemic uncertainties related to earthquake occurrence, we stress that there are significant epistemic uncertainties in ground motion modelling. We would like to achieve a good balance in highlighting both major challenges in our paper.

Another problem with the lack of connection between sections of the paper is that it does not therefore provide a good analysis of how uncertainties can propagate between analysis of related hazards: thus, for example, probabilistic tsunami hazard analysis contains very large uncertainties due to the uncertainties in the understanding of the occurrence distributions of the causative events (mainly earthquakes, so if PSHA is fatally flawed as Stein et al argue, then so is PTHA). Similarly, connections and feedbacks exist between flood hazards and (rainfall-triggered) landslides and also extreme winds associated with intense rainfall since debris from landslides and wind toppled trees entering rivers during extreme flood events, can exacerbate the flooding by blocking flow under bridges and through narrow channels, causing overbank flooding (see the recent examples from Dominica during Hurricane Maria in September 2017). Such feedbacks again require uncertainties to be propagated from one set of models into others, and this is another challenging area with large epistemic and ontological uncertainties that are not adequately covered in this review.

Response: We agree that epistemic uncertainties for earthquake occurrence propagate into tsunami hazards and risks as well as earthquake triggered secondary hazards and risks (e.g. landslides and liquefaction). In Section 6.2, we mentioned that 'Inappropriate application of seismological theories could result in gross underestimation of earthquake magnitude of mega-thrust subduction earthquakes (Stein et al., 2012; Kagan and Jackson, 2013)'. To be clearer, in the revised paper, we include: 'the criticisms towards PSHA (e.g. Mulargia et al., 2017) are also applicable to PTHA'.

A final topic that needs more consideration is that of how the importance of different types of uncertainty varies according to the practical use to which the modelling approach concerned is put. Although the authors of some sections of the paper do make the distinction between probabilistic and real-time warning approaches to hazard mitigation in their respective sections, the implications of this need to be explored further in terms of how tolerant different mitigation strategies are of different types of uncertainty in hazard estimates. Thus, permanent mitigation strategies (in the sense defined by Day & Fearnley, 2015) are extremely sensitive to uncertainties in probabilistic hazard analyses especially at the high-intensity range where the effectiveness of strategies such as building construction codes are liable to break down; whereas responsive and anticipatory mitigation strategies (e.g. tsunami evacuations and volcanic eruption warnings, respectively) are less sensitive to probabilistic uncertainties but are highly dependent on accurate and timely detection and quantification of specific hazard events.

[reference: Day, S.J., & Fearnley, C. (2015). A classification of mitigation strategies for natural hazards: implications for the understanding of interactions between mitigation strategies. Natural Hazards, 79(2), 1219-1238.]

Response: Thank you for the suggestion. We agree that interaction/interference of different mitigation measures is important. This should be applicable to the majority of natural hazards discussed in the paper. Accordingly, we cite Day and Fearnley (2015) in Introduction by mentioning that '*We recognise that different types of hazard mitigation strategy might have different sensitivities to the treatment of epistemic uncertainties*'.

Overall, the paper contains some useful material but it is not properly examined or brought together (which is the whole point of "review" or "personal perspective" papers) so I do not think that it is suitable for publication in its present form and needs further substantial revision and re-review.

Response: While we appreciate that the manuscript is long and can do with some condensing (which we will do), we nonetheless have to stress that this manuscript is part of a 2-part paper. We felt that a two-part paper is needed given the breadth of natural hazard areas discussed. The second paper contains this personal perspective and discussion that the reviewer asks for. We therefore cannot also include this discussion again in paper 1. We will try to make this connection clearer though.

---

## Author Comment (AC2) · 17 Apr 2018

Please see document posted in reply to Editor for all responses
* * *

---

## Author Comment (AC3) · 17 Apr 2018

Please see document posted in reply to Editor for all responses
* * *

---

## Author Response (AR2)

Response to final comments from the Editor.

We have made some final changes (and included the reference that should have been included already) to take account of Richard's comments. We are not sure that we anywhere said that probabilities could only be used to represent aleatory variables (we think a quick search on aleatory in the document will show this); nor do we rule out treating epistemic uncertainties as probabilities (this is often the outcome in the expert elicitations that are discussed in both parts of the paper). What we did say is that uncertain variables in natural hazard assessments are very often treated as if they are aleatory when that may not be appropriate. We have modified the abstract to try and make this clearer.